# Generative Subspace Adversarial Active Learning for Outlier Detection in Multiple Views of High-dimensional Tabular Data

## Abstract

Outlier detection in high-dimensional tabular data is an important task in data mining, essential for many downstream tasks and applications. Existing unsupervised outlier detection algorithms face one or more problems, including inlier assumption (IA), curse of dimensionality (CD), and multiple views (MV). To address these issues, we introduce Generative Subspace Adversarial Active Learning (GSAAL), a novel approach that uses a Generative Adversarial Network with multiple adversaries. These adversaries learn the marginal class probability functions over different data subspaces, while a single generator in the full space models the entire distribution of the inlier class. GSAAL is specifically designed to address the MV limitation while also handling the IA and CD, making it the only method to address all three. We provide a mathematical formulation of MV, theoretical guarantees for the training, and scalability analysis for GSAAL. Our extensive experiments demonstrate the effectiveness and scalability of GSAAL, highlighting its superior performance compared to other popular OD methods, especially in MV scenarios.

## 1   Introduction

Outlier detection (OD), a fundamental and widely recognized issue in data mining, involves the identification of anomalous or deviating data points within a dataset. Outliers are typically defined as low-probability occurrences within a population [41, 19]. In the absence of access to the true probability distribution of the data points, OD algorithms rely on constructing a scoring function. Points with higher scores are more likely to be outliers. Existing unsupervised OD algorithms have one or more of the following problems, in high-dimensional tabular data scenarios.

- *The inlier assumption* (IA): OD algorithms often make assumptions about what constitutes an inlier, which can be challenging to verify and validate [30].
- *The curse of dimensionality* (CD): As the dimensionality of data increases, the challenge of identifying outliers intensifies, decreasing the effectiveness of certain OD algorithms [2]
- *Multiple Views* (MV): Outliers are often only visible in certain "views" of the data and are hidden in the full space of original features [31]

We now explain these problems one by one.

*The inlier assumption* poses a challenge to algorithms that assume a standard profile of the inlier data. For example, angle-based algorithms like ABOD [24] assume that inliers have other inliers at all angles. Similarly, neighbor-based algorithms like kNN [34] assume that inliers have other neighboring points nearby. These assumptions influence the scoring as it measures the degree to which a sample deviates from this assumed norm. Consequently, the performance of these algorithms

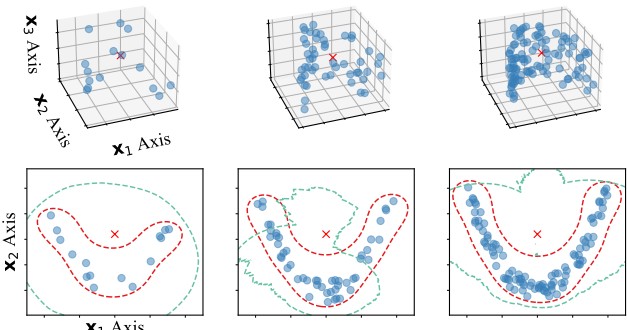

Figure 1: Scatterplots of the dataset from example 1.

may degrade if these assumptions do not hold [30]. This means that a general OD method should not make any inlier assumptions.

*The curse of dimensionality* [2] refers to the decrease in the relative proximity of data points as the number of dimensions increases. Simply put, with high dimensionality, the distance between any pair of points becomes similar, regardless of whether none, one, or both of the points in a pair are outliers. This is particularly problematic for OD algorithms that rely on distances or on identifying neighbors to detect outliers, such as density- (e.g., LOF [3]), neighbor- (e.g., kNN [34]), and cluster-based (e.g., SVDD [1, Chapter 2]) OD algorithms.

*Multiple Views* refers to the phenomenon that certain complex correlations between features are only observable in some feature subspaces [31]. As detailed in [1], this occurs when the dataset contains additional irrelevant features, making some outliers only detectable in certain subspaces. In scenarios where multiple subspaces contain different interesting structures, this problem is exacerbated. It then becomes increasingly difficult to explain the variability of a data point based solely on its behavior in a single subspace [23]. This problem can occur regardless of the dimensionality of the dataset if the number of points is insufficient to capture a complex correlation structure.

The following example illustrates the three problems described above

**Example 1** (Effect of MV, IA and CD). *Consider the random variables $\mathbf{x}_1$, $\mathbf{x}_2$ and $\mathbf{x}_3$, where $\mathbf{x}_1$ and $\mathbf{x}_2$ are highly correlated and $\mathbf{x}_3$ is Gaussian noise. Figure 1 plots datasets with 20, 100 and 1000 realizations of $(\mathbf{x}_1, \mathbf{x}_2, \mathbf{x}_3)$. It also contains the classification boundaries from both a locality-based method (green) and a cluster-based method (red) in the subspace. The cluster-based detector fitted in the full 3D space fails to detect the outlier shown in the figure (red cross). However, the outlier is always detected in the 2D subspace, as we can see. Once we increase the number of samples over $n = 1000$, the cluster-based method detects the outlier in the full space (MV). On the contrary, the locality-based method could not detect the outlier in any tested scenario (MV + IA). If we increase the dimensionality by adding more features consisting of noise, no method can detect the outlier in the full space (MV + IA + CD).*

We are interested in tackling outlier detection whenever a population exhibits MV, like [31, 23, 25] and as showcased in [1]. Particularly, the goal of this paper is to propose the first outlier detection method that explicitly addresses IA, CD, and MV simultaneously.

As we will explain in the next section, we build on Generative Adversarial Active Learning (GAAL) [44], a widely used approach for outlier detection [30, 17, 39]. It involves training a Generative Adversarial Network (GAN) to mimic the distribution of outlier data, and it enhances the discriminator's performance through active learning [38], leveraging the GAN's data generation capability. GAAL methods avoid IA [30] and use the multi-layered structure of the GAN to overcome the curse of dimensionality [33]. However, they often miss important subspaces, leading to MV.

**Challenges.** Training multiple GAN-based models in individual subspaces is not trivial. (1) The joint training of generators and discriminators in GANs requires careful monitoring to determine the optimal stopping point, a task that becomes daunting for large ensembles. (2) The generation of difficult-to-detect points in a subspace remains hard [40]. (3) While several authors have proposed

Table 1: Families of OD methods with the limitations they address.

| Type | IA | CD | MV |
|------|:--:|:--:|:--:|
| Classical | ✗ | ✗ | ✗ |
| Subspace | ✗ | ✓ | ✓ |
| Generative w/ uniform distribution | ✓ | ✗ | ✗ |
| Generative w/ param. distribution | ✗ | ✓ | ✗ |
| Generative w/ subspace behavior | ✗ | ✓ | ✓ |
| GAAL | ✓ | ✓ | ✗ |
| **GSAAL** (Our method) | ✓ | ✓ | ✓ |

multi-adversarial architectures for GANs [11, 5], none of them address adversaries tailored to subspaces composed of feature subsets. Furthermore, these methods may not be suitable for GAAL since they do not have convergence guarantees for detectors, as we will explain.

**Contributions.** (1) We propose GSAAL (Generative Subspace Adversarial Active Learning), a novel GAAL method that uses multiple adversaries to learn the marginal inlier probability functions in different data subspaces. Each adversary focuses on a single subspace. Simultaneously, we train a single generator in the full space to approximate the entire distribution of the inlier class. All networks are trained end-to-end, avoiding the ensembling problem. (2) To our knowledge, we give the first mathematical formulation of the "multiple views" problem. We used it to show the ability of GSAAL to mitigate the MV problem. (3) We formulate the novel optimization problem for GSAAL and give convergence guarantees of each discriminator to the marginal distribution of its respective subspace. We also analyze the worst-case complexity of the method. (4) In extensive experiments we compare GSAAL with multiple competitors. GSAAL was the only method capable of consistently detecting anomalous data under MV. Furthermore, on 22 popular benchmark datasets for the one-class classification task, GSAAL demonstrated SOTA-level performance and was orders of magnitude faster in inference than its best competitors. (5) Our code is publicly available.[1]

Paper outline: Section 2 reviews related work, Section 3 contains the theoretical results for our method, Section 4 features our experimental results, and Section 5 concludes and addresses limitations.

## 2  Related Work

This section is a brief overview of popular unsupervised outlier detection methods for tabular data related to our approach. We categorize them based on their ability to address the specific limitations outlined above. Table 1 is a comparative summary. Further comments about OD in other data types can be found in the appendix.

**Classical Methods**  Conventional outlier detection approaches, such as distance-based strategies like LOF and KNN, angle-based techniques like ABOD, and cluster-based methods like SVDD, rely on specific assumptions on the behavior of inlier data. They use a scoring function to measure deviations from this assumed norm. These methods face the *inlier assumption* limitation by definition. For example, local methods that assume isolated outliers fail when several outlying samples fall together. In addition, many classical methods, which rely on measuring distances, are susceptible to the *curse of dimensionality*. Both limitations impair the effectiveness of these methods [30].

**Subspace Methods**  Subspace-based methods [25] operate in lower-dimensional subspaces formed by subsets of features. They effectively counteract the curse of dimensionality by focusing on identifying so-called "subspace outliers" [22]. These outliers, which are prevalent in high-dimensional datasets with many correlated features, are often elusive to conventional non-subspace methods [29, 31]. However, existing subspace methods inherently operate on specific assumptions on the nature of anomalies in each subspace they explore, and thus face the *inlier assumption* limitation.

**Generative Methods**  A common strategy to mitigate the IA and CD limitations is to reframe the task as a classification task using self-supervision. A prevalent self-supervised technique, particularly

---

[1]https://anonymous.4open.science/r/GSAAL-8D6E

for tabular data, is the generation of artificial outliers [13, 30]. This method involves distinguishing between actual training data and artificially generated data drawn from a predetermined "reference distribution". [21] showed that by approximating the class probability of being a real sample, one approximates the probability function of being an inlier. One then uses this approximation as a scoring function [30]. However, it is not easy to find the right reference distribution, and a poor choice can affect OD by much [21].

A first approach to this challenge proposed the use of naïve reference distributions by uniformly generating data in the space. This approach showed promising results in low-dimensional spaces but failed in high dimensions due to the curse of dimensionality [21]. Other approaches, such as assuming parametric distributions for inlier data [1, Chapter 2] or directly generating in susbpaces [12], can avoid CD when the parametric assumptions are met. Methods that generate in the subspaces can model the subspace behavior, additionally tackling the MV limitation. However, these last two approaches do not address the IA limitation, as they make specific assumptions about the behavior of the inlier data.

**Generative Adversarial Active Learning**   According to [21], the closer the reference distribution is to the inlier distribution, the better the final approximation to the inlier probability function will be. Hence, recent developments in generative methods have focused on learning the reference distribution in conjunction with the classifier. A key approach is the use of Generative Adversarial Networks (GANs), where the generator converges to the inlier distribution [15]. The most common approaches for this are GAAL-based methods [30, 17, 39]. These methods differentiate themselves from other GANs for OD by training the detectors using active learning after normal convergence of the GAN [36, 10]. The architecture of GAAL inherently addresses the curse of dimensionality, as GANs can incorporate layers designed to manage high-dimensional data [33]. In practice, GAAL-based methods outperformed all their competitors in their original work. However, they overlook the behavior of the data in subspaces and therefore may be susceptible to MV.

Our method, GSAAL, incorporates several subspace-focused detectors into GAAL. These detectors approximate the marginal inlier probability functions of their subspaces. Thus, GSAAL effectively addresses MV while inheriting GAAL's ability to overcome IA and CD limitations.

# 3   Our Method: GSAAL

We first formalize the notion of data exhibiting multiple views. We then use it to design our outlier detection method, GSAAL, and give convergence guarantees. Finally, we derive the runtime complexity of GSAAL. All the proofs and extra derivations can be found in the technical appendix.

## 3.1   Multiple Views

Several authors [1, 31, 23, 25, 29] have observed that at times the variability of the data can only be explained from its behavior in some subspaces. Researchers variably call this problem "the subspace problem" [1, 25] or "multiple views of the data" [22, 31]. Previous research has largely focused on practical scenarios, leaving aside the need for a formal definition. In response, we propose a unifying definition of "multiple views" that provides a foundation for developing methods to address this challenge effectively.

The problem "multiple views" of data (MV) arises from two different effects. First, it involves the ability to understand the behavior of a random vector $\mathbf{x}$ by examining lower-dimensional subsets of its components $(\mathbf{x}_1, \ldots, \mathbf{x}_d)$. Second, it stems from the challenge of insufficient data to obtain an effective scoring function in the full space of $\mathbf{x}$. As Example 1 shows, combining these two effects obscures the behavior of the data in the full space. Hence, methods not considering subspaces when building their scoring function may have issues detecting outliers under MV. The next definition formalizes the first effect.

**Definition 1** (myopic distribution)**.** *Consider a random vector* $\mathbf{x} : \Omega \longrightarrow \mathbb{R}^d$ *and* $Diag_{d \times d}(\{0, 1\})$, *the set of diagonal binary matrices without the identity. If there exists a random matrix* $\mathbf{u} : \Omega \longrightarrow Diag_{d \times d}(\{0, 1\})$, *such that*

$$p_{\mathbf{x}}(x) = p_{\mathbf{ux}}(ux) \text{ for almost all } x, \tag{1}$$

*we say that the distribution of* $\mathbf{x}$ *is* myopic to the views of $\mathbf{u}$*. Here,* $x$ *and* $ux$ *are realizations of* $\mathbf{x}$ *and* $\mathbf{ux}$, *and* $p_{\mathbf{x}}$ *and* $p_{\mathbf{ux}}$ *are the pdfs of* $\mathbf{x}$ *and* $\mathbf{ux}$.

It is clear that, under MV, using $p_{\mathbf{ux}}$ to build a scoring function instead of $p_{\mathbf{x}}$ mitigates the effects. This comes as the subspaces selected by $\mathbf{u}$ are smaller in dimensionality. Hence it should take fewer samples to approximate the pdf of $\mathbf{ux}$. The difficulty is that it is not yet clear how to approximate $p_{\mathbf{ux}}$. The following proposition elaborates on a way to do so. It states that by averaging a collection of marginal distributions of $\mathbf{x}$ in the subspaces given by realizations of $\mathbf{u}$, one can approximate the distribution of $p_{\mathbf{ux}}$.

**Proposition 1.** *Let $\mathbf{x}$ and $\mathbf{u}$ be as before with $p_{\mathbf{x}}$ myopic to the views of $\mathbf{u}$. Consider a set of independent realizations of $\mathbf{u}$: $\{u_i\}_{i=1}^{k}$. Then $\frac{1}{k}\sum_i p_{u_i\mathbf{x}}(u_i x)$ is an unbiased statistic for $p_{\mathbf{ux}}(ux)$.*

MV appears when there is a lack of data, and its distribution is myopic. To improve OD under MV, one can exploit the distribution myopicity to model $\mathbf{x}$ in the subspaces, where less data is sufficient. Proposition 1 gives us a way to do so, by approximating $p_{\mathbf{ux}}$. In this way, under myopicity, this also approximates $p_{\mathbf{x}}$, avoiding MV. Our method, GSAAL, exploits these derivations, as we explain next.

## 3.2 GSAAL

GAAL methods tackle IA by being agnostic to outlier definition and mitigate CD through the use of multilayer neural networks [30, 28, 33]. GAAL methods have two steps:

1. *Training of the GAN.* Train the GAN consisting of one generator $\mathcal{G}$ and one detector $\mathcal{D}$ using the usual min-max optimization problem as in [15].
2. *Training of the detector through active learning.* After convergence, $\mathcal{G}$ is fixed, and $\mathcal{D}$ continues to train. This last step is an active learning procedure with [44]. Following [21], $\mathcal{D}(x)$ now approximates the pdf of the training data $p_{\mathbf{x}}$.

After Step 2, the detector converges to $p_{\mathbf{x}}$. However, our goal is to approximate $p_{\mathbf{x}}$ by exploiting a supposed myopicity of the distribution. We extend GAAL methods to also address MV in what follows. The following theorem adapts the objective function of the GAN to the subspace case and gives guarantees that the detectors converge to the marginal pdfs used in Proposition 1:

**Theorem 1.** *Consider $\mathbf{x}$ and $\mathbf{u}$ as in the previous definition, with $x$ a realization of $\mathbf{x}$ and $\{u_i\}_i$ a set of realizations of $\mathbf{u}$. Consider a generator $\mathcal{G} : z \in Z \longmapsto \mathcal{G}(z) \in \mathbb{R}^d$ and $\{\mathcal{D}_i\}$, $i = 1, \ldots, k$, a set of detectors such as $\mathcal{D}_i : u_i x \in S_i \subset \mathbb{R}^d \longmapsto \mathcal{D}_i(u_i x) \in [0,1]$. $Z$ is an arbitrary noise space where $\mathcal{G}$ randomly samples from. Consider the following optimization problem*

$$
\begin{aligned}
\min_{\mathcal{G}} \max_{\mathcal{D}_i,\,\forall i} \sum_i V(\mathcal{G}, \mathcal{D}_i) = \\
\min_{\mathcal{G}} \max_{\mathcal{D}_i,\,\forall i} \sum_i \mathbb{E}_{u_i\mathbf{x}} \log \mathcal{D}_i(u_i x) + \mathbb{E}_z \log\left(1 - \mathcal{D}_i\left(u_i \mathcal{G}(z)\right)\right),
\end{aligned}
\tag{2}
$$

*where each addend $V(\mathcal{G}, \mathcal{D}_i)$ is the binary cross entropy in each subspace. Under these conditions, the following holds:*

- *i) Each detector in optimum is $\mathcal{D}_i^*(u_i x) = \frac{1}{2}, \forall x$. Thus, in optimum $V(\mathcal{G}, \mathcal{D}_i) = -\log(4), \forall i$.*
- *ii) Each individual $\mathcal{D}_i$ converges to $\mathcal{D}_i^*(u_i x) = p_{u_i x}(u_i x)$ after trained in Step 2 of a GAAL method.*
- *iii) $\mathcal{D}^*(x) = \frac{1}{k}\sum_{i=1}^{k} \mathcal{D}_i^*(u_i\mathbf{x})$ approximates $p_{\mathbf{ux}}(ux)$. If $p_{\mathbf{x}}$ is myopic, $\mathcal{D}^*(x)$ also approximates $p_{\mathbf{x}}(x)$.*

Using Theorem 1 we can extend the GAAL methods to the subspace case:

1. *Training the GAN.* Train a GAN with one generator $\mathcal{G}$ and multiple detectors $\{\mathcal{D}_i\}$ with Equation (2) as the objective function. The training of each detector stops when the loss reaches its value with the optimum in Statement $(i)$.
2. *Training of the $k$ detectors by active learning.* Train each $\mathcal{D}_i$ as in Step 2 of a regular GAAL method using $\mathcal{G}$. By Statement $(ii)$ of the Theorem, each $\mathcal{D}_i$ will approximate $p_{u_i\mathbf{x}}$. By Statement $(iii)$, $\mathcal{D}(x) = \frac{1}{k}\sum_{i=1}^{k}\mathcal{D}_i(u_i\mathbf{x})$ will approximate $p_{\mathbf{x}}$ under the myopicity of the data.

We call this generalization of GAAL Generative Subspace Adversarial Active Learning (GSAAL). The appendix contains the pseudo-code for GSAAL.

## 3.3 Complexity

In this section, we focus on studying the theoretical complexity of GSAAL. We study both its usability for training and, more importantly, for inference.

**Theorem 2.** *Consider our GSAAL method with generator $\mathcal{G}$ and detectors $\{\mathcal{D}_i\}_{i=1}^{k}$, each with four fully connected hidden layers, $\sqrt{n}$ nodes in the detectors and $d$ in the generator. Let $D$ be the training data for GSAAL, with $n$ data points and $d$ features. Then the following holds:*

     *i) Time complexity of training is $\mathcal{O}(E_D \cdot n \cdot (k \cdot n + d^2))$. $E_D$ is an unknown complexity variable depicting the unique epochs to convergence for the network in dataset $D$.*

     *ii) Time complexity of single sample inference is in $\mathcal{O}(k \cdot n)$, with $k$ the number of detectors used.*

The linear inference times make GSAAL particularly appealing in situations where the model can be trained once for each dataset, like one-class classification. We build on this particular strength in the following section.

# 4 Experiments

This section presents experiments with GSAAL. We will outline the experimental setting, and examine the handling of "multiple views" in GSAAL and other OD methods. We then evaluate GSAAL's performance against various OD methods and investigate its scalability. The appendix includes a study on the sensitivity to the number of detectors, IA experiments, an ablaition study and extra competitors evaluated in the real world datasets. System specifications are included in the appendix.

## 4.1 Experimental Setting

This section has three parts: First, we describe the synthetic and real data for the outlier detection experiments. Then, we describe the configuration of GSAAL. Finally, we present our competitors.

### 4.1.1 Datasets

**Synthetic.**  We constructed synthetic datasets, each containing two correlated features, $\mathbf{x}_1$ and $\mathbf{x}_2$, along with 58 independent features $\mathbf{x}_j$, $j = 3, \ldots, 60$ consisting of Gaussian noise. This approach simulates datasets that exhibit the MV property by adding irrelevant features into a pair of highly correlated variables. We detail the methodology and all correlation patterns in the technical appendix.

**Real.**  We selected 22 real-world tabular datasets for our experiments from [19]. The selection criteria included datasets with less than 10,000 data points, more than 10 outliers, and more than 15 features, focusing on high-dimensional data while keeping the runtime (of competing OD methods) tractable. Table 2a contains the summary of the datasets. For datasets with multiple versions, we chose the first in alphanumeric order. Details about each dataset are available in the original source [19].

### 4.1.2 Network Settings

**Structure.**  Unless stated otherwise, GSAAL uses the following network architecture. It consists of four fully connected layers with ReLu activation functions used in the generator and the detectors. Each layer in $k = 2\sqrt{d}$ detectors has $\sqrt{n}$ nodes, where $n$ and $d$ are the number of data points and features in the training set, respectively. This configuration ensures linear inference time. The generator has $d$ nodes in each layer, a standard in GAAL approaches, which ensures polynomial training times. We assumed $\mathbf{u}$ to be distributed uniformly across all subspaces. Therefore, we obtained each subspace for the detectors by drawing uniformly from the set of all subspaces.

**Training.**  Like other GAAL methods [30, 44], we train the generator $\mathcal{G}$ together with all the detectors $\mathcal{D}_i$ until the loss of $\mathcal{G}$ stabilizes. Then we train each detector $\mathcal{D}_i$ until convergence with $\mathcal{G}$ fixed. To automate this process, we introduce an early stopping criterion: Training stops when a detector's loss approaches the theoretical optimum ($-\log(4)$), see statement $(ii)$ of Theorem 1. For consistency across experiments, training parameters remain fixed unless otherwise noted. Specifically,

Table 2: Real-world datasets and Competitors

(a) Real-world datasets converted to tabular if needed

| Dataset | Category | Dataset | Category |
|---|---|---|---|
| 20news | Text | MNIST | Image |
| Annthyroid | Health | MVTec | Text |
| Arrhythmia | Cardiology | Optdigits | Image |
| Cardiot.. | Cardiology | Satellite | Astronomy |
| CIFAR10 | Image | Satimage-2 | Astronomy |
| F-MNIST | Image | SpamBase | Document |
| Fault | Industrial | Speech | Linguistics |
| InternetAds | Image | SVHN | Image |
| Ionosphere | Weather | Waveform | Elect. Eng. |
| Landsat | Astronomy | WPBC | Oncology |
| Letter | Image | Hepatitis | Health |

(b) Competitors

| Type | Competitors |
|---|---|
| Classical | kNN, LOF |
| | ABOD, OCSVM w/ `rbf` |
| Subspace | IForest, SOD |
| Gen., uniform dist. | NA (see the text) |
| Gen., parametric dist. | GMM |
| Gen., subspace behavior | NA (see the text) |
| GAAL | MO-GAAL |

252 the learning rates of the detectors and the generator are 0.01 and 0.001, respectively. We use minibatch
253 gradient descent [14] optimization, with a batch size of 500.

### 4.1.3 Competitors

255 We selected popular and accessible methods from each category, as summarized in Table 2b, guided
256 by related work. We excluded generative methods with uniform distributions because they prove
257 ineffective for large datasets [21]. We could not include a generative method with subspace behavior
258 due to operational issues with the most relevant method in this class, [12], caused by its outdated
259 repository. We used the recommended parameters for all methods, as usual in OD [19].

260 We used the `pyod` [43] library to access all competitors except MO-GAAL. We used MO-GAAL
261 from its original source and implemented our method GSAAL in `keras` [6].

## 4.2 Effect of Multiple Views on Outlier Detection

263 To demonstrate the effectiveness of GSAAL under MV, we use synthetic datasets. Visualizing the
264 outlier scoring function in a 60-dimensional space is challenging, so we project it into the $\mathbf{x}_1$-$\mathbf{x}_2$
265 subspace. A method adept at handling MV should have a boundary that accurately reflects the $\mathbf{x}_1$ and
266 $\mathbf{x}_2$ dependency structure. We first generate a synthetic dataset $D^{\text{synth}}$ as described in section 4.1.1
267 and train the OD model. Using this model, we compute the scores for the points $(x_1, x_2, 0, \ldots, 0)$
268 and visualize the level curves on the $\mathbf{x}_1$-$\mathbf{x}_2$ plane.

269 Figure 2 shows results for selected datasets and competitors, which are detailed in the Appendix. It
270 shows the level curves and decision boundaries (dashed lines) of the methods. Notably, our model
271 effectively detects correlations in the right subspace. To quantify this, we generated outliers in the
272 subspace of interest and extra inliers. We tested the one-class classification performance of each
273 method in 10 different MV datasets. On average, GSAAL managed to obtain 0.70 AUC, while the
274 second-best performer (IForest) did not surpass a random classifier —0.49 AUC. All results and
275 further details can be found in section B.2 in the appendix.

## 4.3 One-class Classification

277 This section evaluates GSAAL on a one-class classification task [37]. First, we study the effectiveness
278 of GSAAL on real data. Then, we investigate the scalability of GSAAL in practical scenarios.

### 4.3.1 Real-world Performance

280 We perform the outlier detection experiments on real datasets. Specifically, we take on the task of
281 one-class classification, where the goal is to detect outliers by training only on a collection of inliers
282 [19]. To evaluate the performance of OD methods, we use AUC as it is robust to test data imbalance,
283 a common issue in OD tasks . The procedure is as follows:

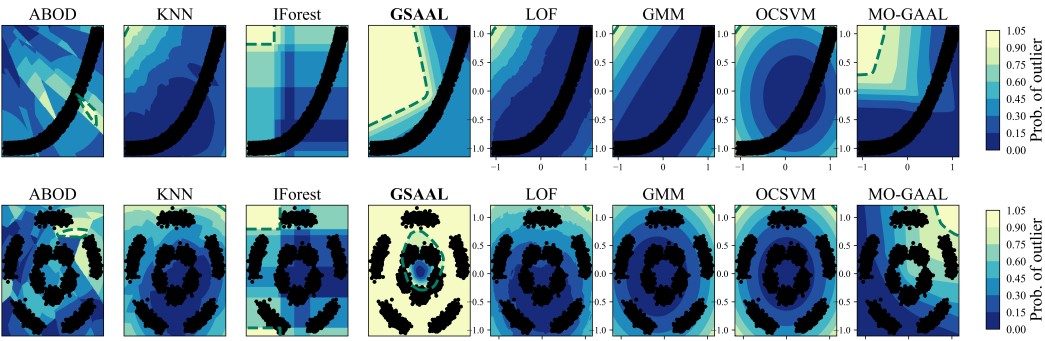

Figure 2: GSAAL finds classification boundaries for datasets banana and star under MV.

Table 3: Results of the Conover-Iman test for pairwise comparisons of the rankings.

| Method | ABOD | **GSAAL** | GMM | IForest | KNN | LOF | MO GAAL | OCSVM | SOD |
|--------|------|-----------|-----|---------|-----|-----|---------|-------|-----|
| ABOD | = | | ++ | ++ | | | ++ | ++ | ++ |
| **GSAAL** | | = | ++ | ++ | | + | ++ | ++ | ++ |
| GMM | −− | −− | = | ++ | −− | −− | | ++ | ++ |
| IForest | −− | −− | −− | = | −− | | ++ | | ++ |
| KNN | | | ++ | ++ | = | | ++ | | ++ |
| LOF | | − | ++ | | | = | ++ | + | ++ |
| MO GAAL | −− | −− | | −− | −− | −− | = | | ++ |
| OCSVM | −− | −− | −− | | | | − | = | ++ |
| SOD | −− | −− | −− | −− | −− | −− | −− | −− | = |

1. Split the dataset $D$ into a training set $D^{\text{train}}$ containing $80\%$ of the inliers from $D$, and a test set $D^{\text{test}}$ containing the remaining inliers and all outliers.
2. Train an outlier detection model with $D^{\text{train}}$ and evaluate its performance on $D^{\text{test}}$ with ROC AUC.

To save space, we moved the detailed AUC results to the appendix; showing that GSAAL obtained the lowest median rank —see Figure 10 in the appendix. Although other subspace methods tend to perform better with irrelevant attributes [29, 25], they did not outperform classical OD methods on average in our experiments. Notably, ABOD, the second-best method in our experiments, performed poorly in the MV tests (Section 4.2).

For statistical comparisons, we use the Conover-Iman post hoc test for pairwise comparisons between multiple populations [7]. It is superior to the Nemenyi test due to its improved type I error boundings [8]. Conover-Iman test requires a preliminary positive result from a multiple population comparison test, for which we employ the Kruskal-Wallis test [26].

Table 3 shows the test results. In each cell, '+' indicates that the method in the row has a significantly lower median rank than the method in the column, while '−' indicates a significantly higher median rank. One symbol indicates p-values $\leq 0.15$ and two symbols indicate p-values $\leq 0.05$. A blank indicates no significant difference. The table shows that GSAAL is superior to most of its competitors. Our method does not significantly outperform the classical methods ABOD and kNN. However, these methods struggle to detect structures in subspaces, showing their inadequacy in dealing with the MV limitation, see Section 4.2.

Overall, the results support GSAAL's superiority in outlier detection tasks involving multiple views. Additionally, they establish our method as the leading GAAL option for One-class classification

### 4.3.2 Scalability

In section 3.3, we derived that the inference time of GSAAL scales linearly with the number of training points if the number of detectors $k$ is fixed, while it does not depend on the number of features $d$. This is in contrast to other methods, in particular LOF, KNN, and ABOD, which have quadratic runtimes in $d$ [3, 24]. We now validate this experimentally. The procedure is as follows:

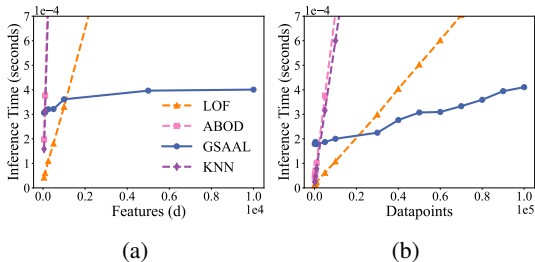

Figure 3: Plots of different performance metrics for scalability

1. Generate datasets $D_{\text{train}}$ and $D_{\text{test}}$ consisting of random points. $|D_{\text{test}}| = 10^6$.
2. Train an OD method using $D_{\text{train}}$ and record the inference time over $D_{\text{test}}$.

Following the result of the sensitivity study in our appendix, we fixed $k = 30$. Figure 3a plots the inference time of a single data point as a function of the number of features when $|D_{train}| = 500$. Figure 3b plots the inference time as a function of the number of points in $D_{\text{train}}$, for a fixed number of 100 features. Both figures confirm our complexity derivations and show that GSAAL is particularly well-suited for large datasets.

## 5 Limitations & Conclusions

### 5.1 Limitations and Future Work

In section 4 we randomly selected subspaces for training the detectors in GSAAL, i.e. we took a uniform distribution of $\mathbf{u}$. This was already sufficient to demonstrate the highly competitive performance of our method. In practice, this assumption seemed to perform well for our experiments. However, GSAAL can work with any subspace search strategy to obtain the distribution of $\mathbf{u}$, for example, the methods exploiting multiple views [23, 22]. We have not included them in this paper due to the lack of an official implementation. In the future, we plan to benchmark various subspace search methods in GSAAL.

Next, GSAAL is limited to tabular data, since the "multiple views" problem has only been observed for this data type. The mathematical formulation of MV in section 3 does not exclude unstructured data. The difficulty lies in identifying good search strategies for $\mathbf{u}$ for non-tabular data, which remains an open question [18]. However, depending on the type of unstructured data, extending GSAAL to work with it is not immediate. Therefore, building a method that exploits the theoretical derivations of GSAAL for structured data is future work.

### 5.2 Conclusions

Unsupervised outlier detection (OD) methods rely on a scoring function to distinguish inliers from outliers, since the true probability function that generated the dataset is usually unavailable in practice. However, they face one or more of the following problems — Inlier Assumption (IA), Curse of Dimensionality (CD), or Multiple Views (MV). In this article, we have proposed the first mathematical formulation of MV, which allows for a better understanding of how to solve this occurrence. Using this formulation, we developed GSAAL, which is the first OD approach that solves MV, CD, and IA. In short, GSAAL is a generative adversarial network with a generator and multiple detectors fitted in the subspaces to find outliers not visible in the full space. In our experiments on 27 different datasets, we demonstrated the usefulness of GSAAL, in particular, its ability to deal with MV and its superior performance on OD tasks with real datasets. In addition, we have shown that GSAAL can scale up to deal with high-dimensional data, which is not the case for our most competent competitors. These results confirm GSAAL's ability to deal with data exhibiting MV and its usability in any practical scenario involving large datasets.

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

# A  Theoretical Appendix

In this appendix, we will include all the proofs of the included theorems and propositions. Additionally, we also extend all non-experimental sections with relevant information for the experimental appendix.

## A.1  Previous Remarks

Before starting to prove our main results, it is important to add a remark about our notation in this article. Whenever we denote $\mathbf{ux}$, we mean the operation resulting in the following vector: $\mathbf{u}(\omega)\mathbf{x}(\omega)$. Thus, $\mathbf{ux}$ is a random vector following its distribution $p_{\mathbf{ux}}$. However, it is important to remark that $ux$, and therefore, also $u_i\mathbf{x}$, does not state the usual matrix-vector multiplication. What we mean by $ux$ is the operation $U \times_M x$, where $U$ stands for the range-complete version of $u$ and $\times_M$ the usual matrix multiplication. This means that whenever we write $ux$ we are considering *the projection of $x$ into the subspace of the features selected in $u$*. This means that $u_i\mathbf{x}$ is the random vector composed of the features selected by $u_i$, and therefore, $p_{u_i\mathbf{x}}(u_ix)$ denotes subsequent marginal pdf of $\mathbf{x}$. We do not state this in the main text as it functionally does not change anything of our derivations, and simply works as a notation. The only important remarks stemming from this fact are the following:

1. $p_{\mathbf{x}}(u_ix) = p_{\mathbf{x}}(\pi_{u_i}(x))$, where $\pi_{u_i}$ denotes the projection of a point $x$ into the subspace of $u_i$. Therefore, we can write $p_{\mathbf{x}}(u_ix) = p_{u_i\mathbf{x}}(u_ix)$.

2. The operator as stated before is not distributive. This is trivial, as given $\mathbf{u}$ a random matrix as in definition 1, $(1_d - \mathbf{u})\mathbf{x}$ is defined properly, as $1_d - \mathbf{u} \in Diag(\{0,1\})$. However, $\mathbf{x} - \mathbf{ux}$ denotes the vector subtraction between two vectors with different dimensionality.

While not important to understand the following proofs and the derivations from the main text, understanding this is crucial for anyone seeking to work with these definitions.

## A.2  Proofs

We will reformulate all of the statements for completion before introducing each proof.

**Proposition 2.** *Let $\mathbf{x}$ and $\mathbf{u}$ be as before with $p_{\mathbf{x}}$ myopic to the views of $\mathbf{u}$. Consider a set of independent realizations of $\mathbf{u}$: $\{u_i\}_{i=1}^k$, a realization of $\mathbf{x}$, $x$, and a realization of $\mathbf{ux}$, $ux$. Then $\frac{1}{k}\sum_i p_{u_i\mathbf{x}}(u_ix)$ is a statistic for $p_{\mathbf{ux}}(ux)$.*

*Proof.* Consider $\mathbf{x}$ and $\mathbf{u}$ as in the statement. Recall the law of total probabilities:
$$p_{\mathbf{ux}}(ux) = \mathbb{E}_{\mathbf{u}}\left(p_{\mathbf{ux}|\mathbf{u}=u'}(ux|u')\right).$$

By taking the definition of $\mathbf{u}$ and the myopicity, it is trivial that:
$$p_{\mathbf{ux}|\mathbf{u}=u'}(ux|u') = p_{u'\mathbf{x}}(u'x)$$

for $u'$ such that $p_{\mathbf{u}}(u') \neq 0$.

Then, by definition of marginal probability and expectation, we have that:

$$p_{\mathbf{ux}}(ux) = \sum_{i=1}^{N} p_{\mathbf{u}}(u_i)p_{u_i\mathbf{x}}(u_ix),$$

as $\mathbf{u}$ is discrete with finite set of occurrences of size $N$. Thus, we can approximate $\sum_{i=1}^{N} p_{\mathbf{u}}(u_i)p_{u_i\mathbf{x}}(u_ix))$ by $\frac{1}{k}\sum_i p_{u_i\mathbf{x}}$ with $u_i$ independent samples of $\mathbf{u}$. $\qquad\square$

**Theorem 3.** *Consider $\mathbf{x}$ and $\mathbf{u}$ as in the previous definition, with $x$ a realization of $\mathbf{x}$ and $\{u_i\}_i$ a set of realizations of $\mathbf{u}$. Consider a generator $\mathcal{G} : z \in Z \longmapsto \mathcal{G}(z) \in \mathbb{R}^d$ and $\{\mathcal{D}_i\}$, $i = 1,\ldots,k$, a set of detectors such as $\mathcal{D}_i : u_ix \in S_i \subset \mathbb{R}^d \longmapsto \mathcal{D}_i(u_ix) \in [0,1]$. $Z$ is an arbitrary noise space where $\mathcal{G}$ randomly samples from. Consider the following objective function*

$$\min_{\mathcal{G}} \max_{\mathcal{D}_i,\,\forall i} \sum_i V(\mathcal{G}, \mathcal{D}_i) =$$
$$\min_{\mathcal{G}} \max_{\mathcal{D}_i,\,\forall i} \sum_i \mathbb{E}_{u_i\mathbf{x}} \log \mathcal{D}_i(u_ix) + \mathbb{E}_z \log\left(1 - \mathcal{D}_i\left(u_i\mathcal{G}(z)\right)\right)$$

(3)

*Under these conditions, the following holds:*

   *i) Each detector's loss in optimum is $V(\mathcal{G}, \mathcal{D}_i^*) = \frac{1}{2}$.*

   *ii) Each individual $\mathcal{D}_i$ converges to $\mathcal{D}_i^*(u_i x) = p_{u_i x}(u_i x)$ after trained in Step 2 of a GAAL method.*

   *iii) $\mathcal{D}^*(x) = \frac{1}{k} \sum_{i=1}^{k} \mathcal{D}_i^*(u_i \mathbf{x})$ approximates $p_{\mathbf{ux}}(ux)$. If $p_{\mathbf{x}}$ is myopic, $\mathcal{D}^*(x)$ also approximates $p_{\mathbf{x}}(x)$.*

*Proof.* This proof will follow mainly the results in [15], adapted for our case. We will first derivative two general results that we are going to use to immediately prove $(i), (ii)$ and $(iii)$. First, consider the objective function

$$\sum_i V(\mathcal{G}, \mathcal{D}_i) = \sum_i \mathbb{E}_{u_i \mathbf{x} \sim p_{u_i \mathbf{x}}} \log(\mathcal{D}_i(u_i x)) +$$
$$\mathbb{E}_{\mathbf{z} \sim p_{\mathbf{z}}}(1 - \log(\mathcal{D}_i(u_i \mathcal{G}(z)))),$$

where $\mathbf{z}$ is the random vector used by $\mathcal{G}$ to sample from the noise space $Z$. We will write $\mathbb{E}_{\mathbf{x}}, \mathbb{E}_{\mathbf{z}}$ and $\mathbb{E}_{u_i \mathbf{x}}$ instead of $\mathbb{E}_{\mathbf{x} \sim p_{\mathbf{x}}}, \mathbb{E}_{\mathbf{z} \sim p_{\mathbf{z}}}$ and $\mathbb{E}_{u_i \mathbf{x} \sim p_{u_i \mathbf{x}}}$ as an abuse of notation.

The problem is, then, to optimize:

$$\min_{\mathcal{G}} \max_{\mathcal{D}_i, \, \forall i} \sum_i V(\mathcal{G}, \mathcal{D}_i). \tag{4}$$

Fixing $\mathcal{G}$ and maximizing for all $\mathcal{D}_i$, each detector individually maximizes $V(\mathcal{G}, \mathcal{D}_i)$. Let us try to obtain the optimal of each $\mathcal{D}_i$ with a fixed $\mathcal{G}$. First, we write:

$$V(\mathcal{G}, \mathcal{D}_i) = \int_{u_i x} p_{u_i \mathbf{x}}(u_i x) \log \mathcal{D}_i(u_i x) du_i x +$$
$$\int_z p_{\mathbf{z}}(z) \log(1 - \mathcal{D}_i(u_i \mathcal{G}(z))) dz.$$

As $\mathcal{G}$ uses $\mathbf{z}$ to sample from its sample distribution $p_{\mathcal{G}}(x)$, we can rewrite the second addent, like in [15], as:

$$V(\mathcal{G}, \mathcal{D}_i) = \int_{u_i x} p_{u_i \mathbf{x}}(u_i x) \log \mathcal{D}_i(u_i x) du_i x +$$
$$\int_{u_i x} p_{\mathcal{G}}(u_i x) \log(1 - \mathcal{D}_i(u_i x)) du_i x.$$

Aggregating both integrals, we have a function of the type $f(t) = a \log(t) + b \log(1 - t)$, with $a, b \in \mathbb{R} - \{0\}$. We know that $f(t)$ obtains its optimum in $t = \frac{a}{a+b}$. As $f(t) \in \mathbb{R}^+$, $V(\mathcal{G}, \mathcal{D}_i)$ obtains its optimum for a given $\mathcal{G}$ in:

$$D_i^*(u_i x) = \frac{p_{u_i \mathbf{x}}(u_i x)}{p_{u_i \mathbf{x}}(u_i x) + p_{\mathcal{G}}(u_i x)}. \tag{5}$$

Let us now consider the following function

$$C(\mathcal{G}) = \sum_i \max_{\mathcal{D}_i, \, \forall i} V(\mathcal{G}, \mathcal{D}_i)$$
$$= \sum_i \mathbb{E}_{u_i \mathbf{x}} \log \frac{p_{u_i \mathbf{x}}(u_i x)}{p_{u_i \mathbf{x}}(u_i x) + p_{\mathcal{G}}(u_i x)} + \tag{6}$$
$$\mathbb{E}_{u_i \mathbf{x} \sim p_{\mathcal{G}}} \log \frac{p_{\mathcal{G}}(u_i x)}{p_{u_i \mathbf{x}}(u_i x) + p_{\mathcal{G}}(u_i x)}.$$

This is known in Game Theory as the cost function of player "$\mathcal{G}$" in the null-sum game defined by the $\min \max$ optimization problem. [15] refers to it as the virtual training criterion of the GAN. The adversarial game defined by (4) reaches an equilibrium (and thus, the $\min \max$ problem an optimum) whenever $C(\mathcal{G})$ is minimized. We will study the value of $\mathcal{G}$ in such equilibrium and use it, together with (5), to prove the statements.

Rewriting $C(\mathcal{G})$ it is clear that:

$$C(\mathcal{G}) = \sum_i KL\left(p_{u_i\mathbf{x}(u_ix)}\|\frac{p_{u_i\mathbf{x}}(u_ix) + p_{\mathcal{G}}(u_ix)}{2}\right)$$
$$+ KL\left(p_{\mathcal{G}}(u_ix)\|\frac{p_{u_i\mathbf{x}}(u_ix) + p_{\mathcal{G}}(u_ix)}{2}\right).$$

This expression corresponds to that of a sum of multiple binary cross entropies between a population coming from $p_{u_i\mathbf{x}}$ and from $p_{\mathcal{G}}$ projected by $u_i$. Therefore, as we know, we can rewrite:

$$C(G) = \sum_i 2JSD(p_{u_i\mathbf{x}(u_ix)}\|p_{\mathcal{G}}(u_ix)),$$

with $JSD$ the Jensen-Shannon divergence. Since $JSD(s\|r) \in [0, \log(2))$, it is clear that $C(\mathcal{G})$ obtains its minimum only whenever

$$p_{\mathcal{G}}(u_ix) = p_{u_i\mathbf{x}}(u_ix), \forall\forall x^2; \tag{7}$$

and for all $i \in \{1, \ldots, k\}$.

Knowing $\mathcal{G}$ and $\mathcal{D}_i$ in the optimum for all $i$, we can prove the statements above:

**(i)** As $p_{\mathcal{G}}(u_ix) = p_{u_i\mathbf{x}}(u_ix)$ for almost all $x$, in the optimum of (4), it is immediate that:

$$\mathcal{D}_i(u_ix) = \frac{1}{2},$$

i.e., the detectors cannot differentiate between the real training data and the synthetic data of the generator. If one employs the numerically stable version of each $V(\mathcal{G}, \mathcal{D}_i)$ (equivalent to the numerically stable version of the binary cross entropy [6]), it is trivial to see that

$$V^{\text{stable}}(\mathcal{G}, \mathcal{D}_i) = \log(2).$$

**(ii)** After optimizing (4), training each $D_i$ individually with $\mathcal{G}$ fixed, is the equivalent of building a two-class classifier distinguishing between the artificial class generated by $p_{\mathcal{G}}(u_ix) = p_{u_i\mathbf{x}}(u_ix)$ and the real data coming from $p_{u_i\mathbf{x}}(u_ix)$. By [21], the resulting two-class classifier would be such as:

$$D_i(u_ix) = p_{u_i\mathbf{x}}(u_ix).$$

**(iii)** By proposition 2 and statement $(ii)$, $\frac{1}{k}\sum_i D_i^*(u_ix)$ is an estimator for $p_{\mathbf{ux}}(ux)$. By myopicity, it is also of $p_{\mathbf{x}}(x)$. $\qquad\square$

**Theorem 4.** *Giving our GSAAL method with generator $\mathcal{G}$ and detectors $\{\mathcal{D}_i\}_{i=1}^k$, each with four fully connected hidden layers, $\sqrt{n}$ nodes in the detectors and $d$ in the generator, we obtain that:*

    *i) The training time complexity is bounded with $\mathcal{O}(E_D \cdot n \cdot (k \cdot n + d^2))$, for a dataset $D$ with $n$ training samples and $d$ features. $E_D$ is an unknown complexity variable depicting the unique epochs to convergence for the network in dataset $D$.*

    *ii) The single sample inference time complexity is bounded with $\mathcal{O}(k \cdot n)$, with $k$ the number of detectors used.*

*Proof.* An evaluation of a neural network is composed of two steps, the backpropagation, and the fowardpass steps. While training the network requires both, inference requires only a fowardpass. Therefore, we will first prove $(ii)$ and will build upon it to prove $(i)$.

---
[2]For almost all $x$

**(ii).** GSAAL consists of a generator and $k$ detectors. Single point inference consists of a single fowardpass of all the detectors. We will first prove the general complexity of a fowardpass of a general fully connected 4 layer network and will use it to derive all the other complexities. Let us consider three weight matrices $W_{ji}$, $W_{hj}$ and $W_{lh}$ each between two layers, with $j, i, h$ and $l$ being the number of nodes in each. Therefore, $W_{ji}$ denotes a matrix with $j$ rows and $i$ columns, and so on. Now, let us consider $x_{i1}$ the datapoint after passing the input layer. Lastly, without any loss of generality, consider $f$ to be the activation function for all layers. This way, the forward pass of a single detector can be written as:

$$c_{l1} = f\left(W_{lh}f\left(W_{hj}f\left(W_{ji}x_{i1}\right)\right)\right).$$

We will study the complexity in the first layer and use it to derive the complexity of the others. $A_{j1} = W_{ji}x_{i1}$ is a simple matrix-vector multiplication that we know to be $\mathcal{O}(j \cdot i)$ atmost. Then, as $f$ is an activation function, $f(A_{j1})$ is equivalent to writing $f_{j1} \odot A_{j1}$, with $\odot$ being the element-wise multiplication. Thus, $f\left(W_{ji}x_{i1}\right)$ is:

$$\mathcal{O}(j \cdot i + j) = \mathcal{O}(j \cdot (i+1)) = \mathcal{O}(j \cdot i).$$

Doing this for all layers, we obtain:

$$\mathcal{O}(l \cdot h + k \cdot j + j \cdot i). \tag{8}$$

As all layers have $\sqrt{n}$ nodes,

$$\mathcal{O}(3n) = \mathcal{O}(n).$$

As we have $k$ detectors, the complexity for a fowardpass of all detectors, and thus, for a single sample inference of GSAAL is:

$$\mathcal{O}(k \cdot n).$$

**(i).** A backpropagation step has the same complexity as an inference step on all training samples. As we have $n$ training samples, this then becomes

$$\mathcal{O}(k \cdot n^2)$$

for the detectors. As the training consists of multiple epochs, we will write

$$\mathcal{O}(E_D \cdot k \cdot n^2),$$

with $E_D$ being the number of epochs needed for convergence for the training data set $D$. As the training consists of both backpropagation and fowardpass steps on all training samples, the total training time complexity for all detectors is:

$$\mathcal{O}(E_D \cdot k \cdot n^2 + k \cdot n^2) = \mathcal{O}(E_D \cdot k \cdot n^2).$$

As we also need to consider the generator, we will use equation 8 to derive both steps on the generator. As the generator is also a fully connected 4-layer network, with all layers having $d$ nodes, the complexity for a single fowardpass is:

$$\mathcal{O}(d^2).$$

As during training one generates $n$ samples during each fowardpass:

$$\mathcal{O}(n \cdot d^2).$$

Now, on each backpropagation pass the network calculates the backpropagation error for each generated sample, thus,

$$\mathcal{O}(n \cdot d^2)$$

is also the time complexity for the backpropagation step of the generator. Considering all $E_D$ epochs and both backpropagation and fowardpass steps of the generator and all the detectors, the time complexity of GSAAL's training is:

$$\mathcal{O}(E_D \cdot k \cdot n^2 + E_D \cdot n \cdot d^2) = \mathcal{O}(E_D \cdot n \cdot (k \cdot n + d^2))$$

$\square$

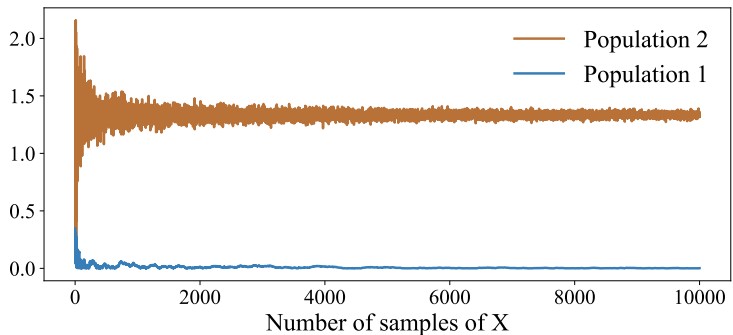

Figure 4: Difference in statistical distance between two populations.

## A.3 Related Work (extension)

**Deep Outlier Detection for other data types.** Outlier detection is also very popular in different data types, especially in unstructured data [42, 16, 36, 35, 32]. Due to the complexity of the data they are used for, deep methods are the main approach employed for this task. The main difference with the other deep methods introduced for tabular data, is that the deep architecture in the later targets mainly CD. For unstructured data types, like images or natural language, is the complexity of the data that drives the architecture. For example, to treat image data, multiple linear layers do not suffice, complex layers like convolutional or residual layers are employed for this [27].

Although popular, most deep methods have limited to no use at all in tabula data in their original articles. However, some have appeared in the literature of tabular data as competitors [36, 35]. We identified the most common for our task in related articles and benchmarks, and included them as an extension of our main experiments in sections B.2 and B.3.

## A.4 Multiple Views (extension)

In this section we extend the derivations in section 3.1 by providing an example of a myopic distribution:

**Example 2** (Myopic distribution). *Consider a* $\mathbf{x}$ *like in example 1. Here, it is clear that* $\mathbf{x_1}, \mathbf{x_2} \perp \mathbf{x_3}$. *Consider, then,* $\mathbf{u}$ *such that:*

$$\mathbf{u} : \{1\} \longrightarrow \{diag(1, 1, 0)\}.$$

*To test whether* $p_{\mathbf{x}}$ *is myopic, we employed a simple test utilizing a statistical distance ($MMD$ with the identity kernel) between* $p_{\mathbf{x}}$ *and* $p_{\mathbf{ux}}$. *This way, if* $M\hat{M}D(p_{\mathbf{x}} \| p_{\mathbf{ux}}) = 0$, *it would be clear that the equality holds. As a control measure, we also calculated the same distance for a different population* $\mathbf{x}'$, *where* $\mathbf{x_3} = \mathbf{x_1^2}$. *We have plotted the results in image 4, where Population 1 refers to* $\mathbf{x}$ *and Population 2 to* $\mathbf{x}'$. *As we can see, we do obtain a positive result in the test of myopicity for* $\mathbf{x}$ *and a negative one for* $\mathbf{x}'$.

## A.5 GSAAL (extension)

We now extend the results from section 3.2 by providing the pseudocode for the training of our method. It is important to consider that, while theorem 3 formulates the optimization problem in terms of the neural networks $\mathcal{G}$ and $\{\mathcal{D}_i\}_i$, in practice this will not be the case. Instead, we will consider the optimization in terms of their weights, $\Theta_{\mathcal{G}}$ and $\Theta_{\mathcal{D}_i}$. Therefore, in practice, the convergence into an equilibrium will be limited by the capacity of the networks themselves [14]. We considered the optimization to follow minibatch-stochastic gradient descent [14]. To consider any other minibatch-gradient method it will suffice to perform the necessary transformations to the gradients.

The pseudocode is located in Algorithm 1. As it is the training for the method, it takes both the parameters for the method and the training. In this case, $epochs$ refers to the total number of epochs we will train in total, while $stop\_epoch$ marks the epoch where we start step 2 of the GAAL training. Lines 1-3 initialize both the detectors in their subspaces and the generator with

**Algorithm 1** GSAAL training

---

**Require:** Data set $D$, Number of Discriminators $\kappa$, $\mathbf{u}$, $epochs$, $stop\_epoch$
1: Initialize Generator $\mathcal{G}$ {#$d$ is the dimensionality of $D$}
2: $\{u_i\}_{i=1}^{\kappa} \leftarrow \text{DRAWFROM}\mathbf{u}(\kappa)$
3: Initialize Discriminators $\{\mathcal{D}_i\}_{i=1}^{\kappa}$ with unique subspaces $\{u_i\}_{i=1}^{\kappa}$
4: **for** $epoch \in \{1, ..., epochs\}$ **do**
5:    **for** $batch \in \{1, ..., batches\}$ **do**
6:       $noise \leftarrow$ Random noise $z^{(1)}, ..., z^{(m)}$ from $Z$
7:       $data \leftarrow$ Draw current batch $x^{(1)}, ..., x^{(m)}$
8:       **for** $j \in \{1...k\}$ **do**
9:          Update $\mathcal{D}_j$ by ascending the stochastic gradient: $\nabla_{\Theta_{\mathcal{D}_j}} \frac{1}{m} \sum_{i=1}^{m} \log(\mathcal{D}_j(u_j x^{(i)})) +$
         $\log(1 - \mathcal{D}_j(u_j \mathcal{G}(z^{(i)})))$
10:       **end for**
11:       **if** $epoch < stop\_epoch$ **then**
12:          Update $\mathcal{G}$ by descending the stochastic gradient: $\nabla_{\Theta_G} \frac{1}{k} \sum_{j=1}^{k} \frac{1}{m} \sum_{i=1}^{m} \log(1 -$
         $\mathcal{D}_j(\mathcal{G}(z^{(i)})))$
13:       **end if**
14:    **end for**
15: **end for**

---

Table 4: Different outliers generated for the experiments.

| Outlier Type | Assumption Description | Outlier Description | $M$ |
|---|---|---|---|
| Local | Assumes that all inliers are located close to other inliers | As a result, outliers are far away from inliers | LOF |
| Angle | Assumes that all inliers have other inliers in all angles from their position | As a result, outliers are not surrounded by other points | ABOD |
| Cluster | Assumes that all inliers form large clusters of data | As a result, outliers are gathered in small clusters | $F_{n, \mu + \varepsilon_i}$ |

random weight matrices $\Theta_{\mathcal{D}_i}$ and $\Theta_{\mathcal{G}}$. Lines 4-13 correspond to the normal GAN training loop across multiple epochs, referred to as step 1 of a GAAL method, if $epoch < stop\_epoch$. Here we proceed with training each detector and the generator using their gradients. Lines 8-10 update each detector by ascending its stochastic gradient, while line 11 updates the generator by descending its stochastic gradient. After the normal GAN training, we start the active learning loop [30] once $epoch \geq stop\_epoch$. The only difference with the regular GAN training is that $\mathcal{G}$ remains fixed, i.e., we do not descend using its gradient. This allows us to additionally train the detectors and, in case of equilibrium of step 1, converge to the desired marginal distributions as derived in theorem 3.

# B   Experimental Appendix

In this section, we will include a supplementary experiment testing the IA condition for completion, the sensibility experiments, and an ablation study. Additionally, we extended both main experimental studies featured in the main text. All of the code for the extra experiments, as well as for all experiments in the main text, can be found in our remote repository[3]. Our experiments used a RTX 3090 GPU and an AMD EPYC 7443p CPU running Python in Ubuntu 22.04.3 LTS. Deep neural network methods were trained on the GPU and inferred on the CPU; shallow methods used only the CPU.

## B.1   Effects of Inlier Assumptions on Outlier Detection

GAAL methodologies are capable of dealing with the inlier assumption by learning the correct inlier distribution $p_{\mathbf{x}}$ without any assumption [30]. While this should also extend to our methodology, we will study experimentally whether this condition holds in practice. To do so, as one cannot identify

---

[3]https://anonymous.4open.science/r/GSAAL-8D6E

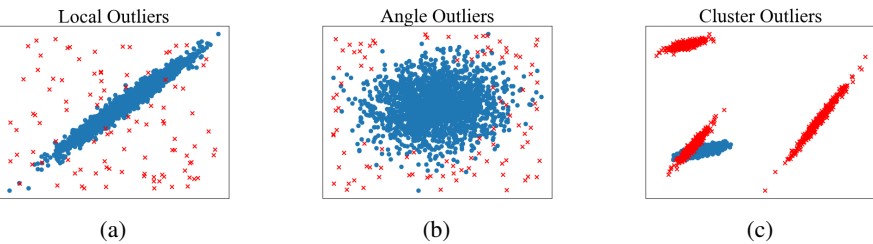

Figure 5: 2D-example of the different types of anomalies we generate using the method summarized in table 4.

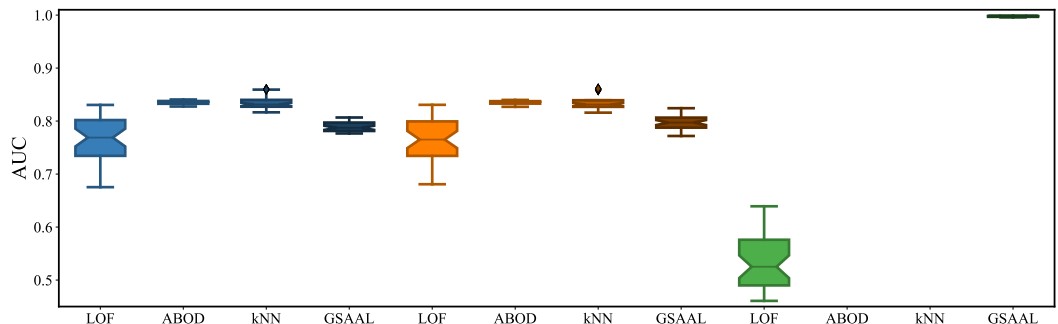

Figure 6: AUCs of the different methods in the IA experiments. From left to right: Local (blue), Angle (orange) and Cluster (green).

beforehand whether a method is going to fail due to IA, we will generate synthetic datasets. This will allow us to generate outliers that we know to follow from a specific IA, ensuring that failure comes from the anomalies themselves. We will include all of the code in the code repository. To generate the synthetic datasets we follow:

1. Generate $D$, a population of 2000 inliers following some distribution $F$ in $\mathbb{R}^{20}$.

2. Select an outlier detection method $M$ with some assumption about the normality of the data and fit it using $D$. We will call such $M$ as the reference model for the generation.

3. Generate 400 outliers by sampling on $\mathbb{R}^{20}$ uniformly and keeping only those points $o$ such that $M(o) = 1$ (i.e., they are detected as outliers). We will write $O^D$ to refer to such a collection of points.

4. Repeat step 3 10 times, to obtain $O_1^D, \ldots, O_{10}^D$.

5. Sample out 20% of the points in $D$. The remainder 80% will be stored in $D^{\text{train}}$, and the other 20% in $D_1^{\text{test}}, \ldots, D_{10}^{\text{test}}$ together with each $O_i^D$.

These steps were repeated 4 times with different $F$, to create 4 different training sets and 40 different testing sets, corresponding to a total of 40 different datasets employed per model $M$ selected in step 2. As we used 3 different reference models, we have a total of 120 different datasets employed in this experiment alone. In particular, the models used for this are collected in table 4. The table contains the name of the outlier type, the description of the IA taken to generate them, and a brief description of how the outliers should look. Column $M$ contains the method employed to generate each, these being $LOF$, $ABOD$, and the same inlier distribution as $D$, but with multiple shifted means $\mu_i$ and with a significantly lower amount of points $n$. A visualization of how these outliers would look with 2 features is located in figure 5. To study how different methods behave when detecting these outliers, we have performed the same experiments as in section 4.3, but with these synthetic datasets. Figure 6 gathers all the AUCs of a method in 3 boxplots, one for each outlier type in each training set. Additionally, we grouped all based on the IA and assigned a similar color for all of them. We have done this for the classical OD methods LOF, ABOD, and kNN, besides our method GSAAL. We cropped the image below 0.45 in the $y$ axis as we are not interested in results below a random classifier. As we can see, classical methods seem to correctly detect outliers for

an outlier type that verifies its IA. However, whenever we introduce outliers behaving outside of their IA, the performance hit is significant. Notoriously, it appears that none of them had trouble detecting the *Local* and *Angle* outlier type. regardless of their IA. This can be easily explained by those outliers types being similar, as we can see in figure 5. On the other hand, GSAAL manages to have a significant detection rate regardless of the outlier type.

## B.2 Effects of Multiple Views on Outlier Detection (extension)

In this section, we will include a brief description of the generation process for the datasets used in section 4.2. We will also perform the same experiment as in section 4.2 for all methods showcased in the main text and additional datasets. The datasets were generated by the following formulas:

- *Banana.* Given $\theta \in [0, \pi]$ we have $\mathbf{x} = \sin(\theta) + U(0, 0.1)$ and $\mathbf{y} = \sin(\theta)^3 + U(0, 0.1)$.
- *Spiral.* Given $\theta \in [0, 4\pi]$ and $r \in (0, 1)$, we have $\mathbf{x} = r\cos(\theta) + U(0, 0.1)$ and $\mathbf{y} = r\sin(\theta)$.
- *Star.* Given $\theta \in [0, 2\pi]$ and $r \in \{r \in \mathbb{R} | r = \sin(5\theta); r \geq 0, 1, 0.4\}$, we have $\mathbf{x} = r\cos(\theta) + U(0, 0.1)$ and $\mathbf{y} = r\sin(\theta) + U(0, 0.1)$.
- *Circle.* Given $\theta \in [0, 2\pi]$, we have $\mathbf{x} = \cos(\theta) + U(0, 0.1)$ and $\mathbf{y} = \sin(\theta) + U(0, 0.1)$.
- *L.* Given $x_1 = N(0, 0.1), x_2 = U(0, 5), y_1 = U(-5, 0),$ and $y_2 = N(0, 0.1)$; we have $\mathbf{x} = \text{concat}(x_1, x_2)$ and $\mathbf{y} = \text{concat}(y_1, y_2)$.

We considered $N(0, 0.1)$ to denote a random normal realization with $\mu = 0$ and $\sigma^2 = 0.1$, and $U(a, b)$ to denote a uniform realization in the $[a, b]$ interval.

Figure 7 contains all images from the MV experiment. We employed the default parameters for all methods in this experiments. We did that as those were the employed parameters in our real world experiments. Additonally, the choice of parameter did not impact the outcome of the experiment much. Our remote repository includes extra images for every competitor with multiple parameters for comparison. We do not have any new insight beyond the ones exposed in the main article. Note that we have included all methods but SOD. The reason was that SOD failed to execute for datasets Star, Spiral, and Circle.

Additionally, we added competitors from outside of our related work that will later be used in section B.3. In particular, we employed LUNAR, DIF and DeepSVDD with default parameters. We included extra images in our remote repository with multiple parameters for the deep competitors as well. The method AnoGAN was not included due to it failing in datasets Star, Spiral and Circle. Their results can be seen in Figure 8. As it also happened our main competitors, some of the extra competitors were capable of detecting the data structure in very sparse occasions. However they remained incapable to properly describe a boundary consistently. The only method that was sensible enough in all datasets was GSAAL.

In order to quantify this, we tested the ability of all methods to perform one-class classification in each dataset. As outliers, we used white noise in the $\mathbf{x}_1 - \mathbf{x}_2$ subspace. Additionally, we created two extra datasets greatly different from the rest, *X* and *wave*:

- *X.* Given $x_1 = x_2 = U(-1, 1)$ and $y_1 = x_1 + U(0, 0.1), y_2 = x_2 + U(0, 0.1)$; we have $\mathbf{x} = \text{concat}(x_1, x_2)$ and $\mathbf{y} = \text{concat}(y_1, y_2)$..
- *Wave.* Given $\theta \in [0, 4\pi]$, we have $\mathbf{x} = \theta$ and $\mathbf{y} = \sin(x) + U(0, 0.1)$.

We will also use them as outleirs, for a total of 15 different datasets. We also generated extra inliers in each test set. We gathered the AUC results in Figure 9. As we can see, all other methods struggel to come ahead of the random classifier, marked with a dashed line. The only method well above that is GSAAL.

## B.3 One-class Classification (extension)

As we noted in Section 4, we obtained our benchmark datasets from [19], a benchmark study for One-class classification methods in tabular data. Some of the datasets featured in the study, and also in our experiments, were obtained from embedding image or text data using a pre-trained NN

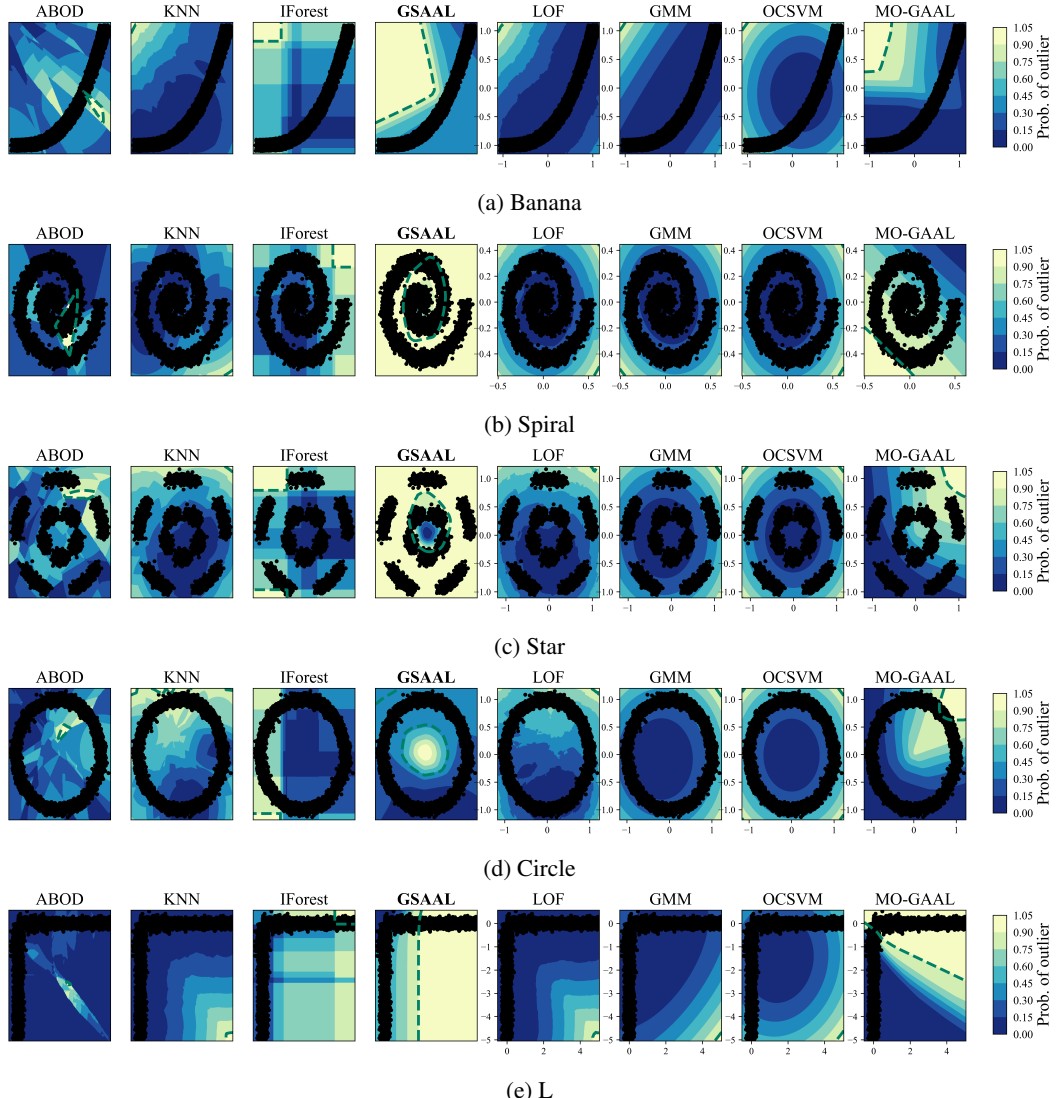

Figure 7: Projected classification boundaries for the datasets in section 4.2 and the extra datasets.

(ResNet [20] and BERT [9], respectively). We shunt the interested reader into [19] for additional information. Additionally, we found discrepancies between the versions of the datasets in the study of [4] and [19]. We utilized the version of those datasets featured in [4] for our experiments due to popularity. This affected the datasets *Arrhythmia, Annthyroid, Cardiotocography, InternetAds, Ionosphere, SpamBase, Waveform, WPBC* and *Hepatitis*. Figure 10 summarizes the ranks from the one-class experiments in section 4.3. Table 5 summarizes the AUC results from our experiments. As mentioned in section A.3, we also included extra methods outside of our related work. Particularly, we added deep versions tailored to image data of previously included methods —DeepSVDD [35] and Deep Isolation Forest [42] (DIF)— and others that extend some types of outlier detectors into image and text data —LUNAR [16], as an extension of Locality-based classical methods, and AnoGAN [36], as an extension of Generative methods. For their parameters, we employed the recommended ones for LUNAR and DIF, and trained the models the same way that the authors did in their articles. As for DeepSVDD and AnoGAN, as they do not have any recommended way of training nor hyperparameters, we performed a grid search for their training parameters and kept the best result. We used all of their official implementations[4]. All deep methods (including MO-GAAL

---

[4]LUNAR and DIF have official implementations by their authors in `pyod` [43].

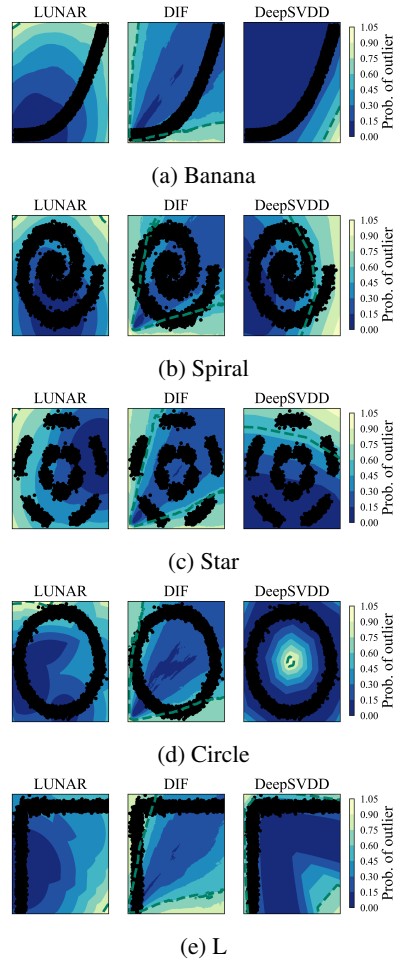

Figure 8: Projected classification boundaries of the competitors outside of our related work.

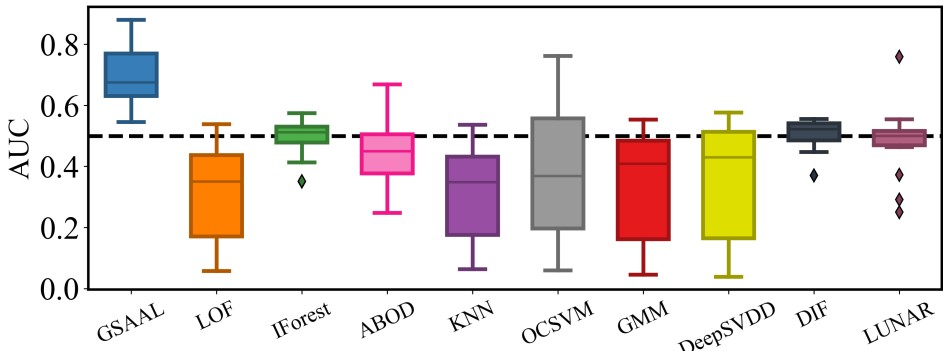

Figure 9: AUC results in the MV datasets.

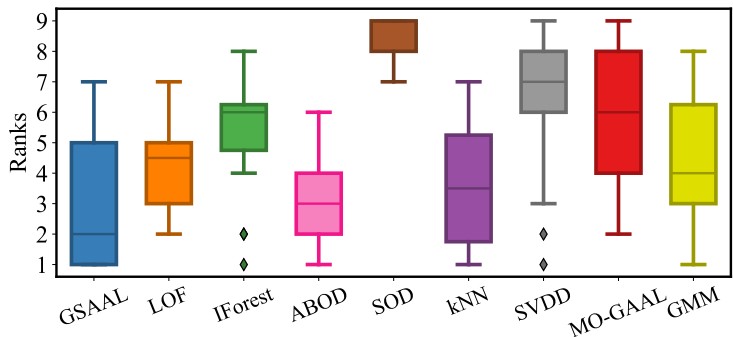

Figure 10: Boxplots of the ranks used for the Conover-Iman experiment in section 4.3.

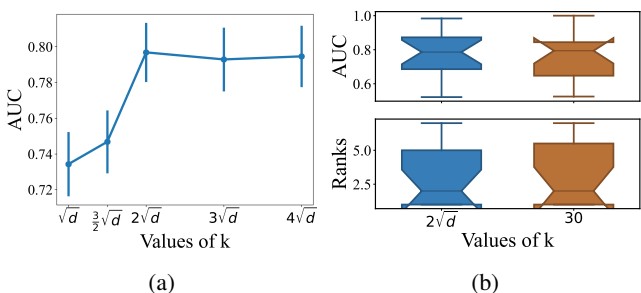

(a)             (b)

Figure 11: Performance of the detector with different values of $k$.

and GSAAL) were trained multiple times with the same train set and their results were averaged to account for initialization.

Additionally, we gathered all extra deep methods and performed the same statistical analysis as in section 4.3. We also included MO GAAL besides GSAAL for completion. SO GAAL, the single generator version of MO GAAL was not included, even if popular in the related literature. The reason is that authors in [30] showed that MO GAAL constantly outperforms SO GAAL in the outlier detection task. Results are included in table 6, gathered after a positive Kruskal-Wallis test. As we can see, GSAAL outperform almost all competitors except LUNAR (the most recent method). However, LUNAR is incapable to detect change in the subspaces as GSAAL does, see section B.2. Therefore, regardless of considering the tabular related work, or the more generalist deep methods, GSAAL still can outperform most competitors in the field. Additionally, for those that GSAAL performs similar to, we showed that we are more sensible to changes in subspaces. This fact makes GSAAL the preferred option for One-class classification under MV.

### B.4 Parameter Sensibility

We now explore the effect of the number of detectors in GSAAL, $k$, by repeating the previous experiments with varying $k$. Figure 11a plots the median AUC for different $k$ values, showing a stabilization at larger $k$. Next, Figure 11b compares the results with a fixed $k = 30$ and the default value $k = 2\sqrt{d}$ used in the previous experiments; there is no large difference in either the AUC or the ranks. We also found that the results in Table 3 remain almost the same if one sets $k = 30$. So we recommend fixing $k = 30$, which makes GSAAL very suitable for high-dimensional data.

### B.5 Ablation study

Lastly, we also performed an ablation study for GSAAL. We identify two critical components in our method, the subspace nature of our detectors, and the multiple detectors used. Table 7 contains a summary of the included features in each considered configuration. We will compare the performance of all the different configurations of GSAAL.

Table 5: AUC of all the methods tested in section 4.3 and extra methods.

| Dataset | GSAAL | LOF | IForest | ABOD | SOD | KNN | SVDD | MO-GAAL | GMM | DeepSVDD | AnoGAN | DIF | LUNAR |
|---|---|---|---|---|---|---|---|---|---|---|---|---|---|
| annthyroid | 0,7681 | 0,6753 | 0,7094 | 0,7008 | 0,5243 | 0,6291 | 0,4611 | 0,5047 | 0,6932 | 0,872 | 0,4038 | 0,6228 | 0,8120 |
| Arrhythmia | 0,7532 | 0,7277 | 0,7695 | 0,7422 | 0,6514 | 0,7334 | 0,7442 | 0,6901 | 0,7296 | 0,7485 | 0,6133 | 0,7904 | 0,7412 |
| Cardiotocography | 0,8727 | 0,8038 | 0,7772 | 0,7956 | 0,3524 | 0,7733 | 0,8351 | 0,7912 | 0,7413 | 0,874 | 0,3248 | 0,5561 | 0,8219 |
| CIFAR10 | 0,7862 | 0,7333 | 0,6853 | 0,7622 | 0,6607 | 0,7493 | 0,7074 | 0,6256 | 0,7462 | 0,6158 | 0,3705 | 0,6542 | 0,7612 |
| FashionMNIST | 0,8001 | 0,8995 | 0,8298 | 0,9009 | 0,7136 | 0,9179 | 0,8130 | 0,7930 | 0,9072 | 0,6981 | 0,7137 | 0,8336 | 0,9093 |
| fault | 0,6726 | 0,6436 | 0,6518 | 0,8019 | 0,5670 | 0,7849 | 0,5651 | 0,6821 | 0,6856 | 0,4972 | 0,4074 | 0,7240 | 0,8047 |
| InternetAds | 0,7809 | 0,8565 | 0,4739 | 0,8600 | 0,3663 | 0,8090 | 0,7063 | 0,7603 | 0,9113 | 0,8411 | 0,5165 | 0,4330 | 0,8036 |
| Ionosphere | 0,9593 | 0,9591 | 0,9377 | 0,9483 | 0,8250 | 0,9825 | 0,8379 | 0,9727 | 0,9644 | 0,967 | 0,8406 | 0,9159 | 0,9234 |
| landsat | 0,5217 | 0,7598 | 0,5927 | 0,7627 | 0,4821 | 0,7726 | 0,4792 | 0,4432 | 0,4998 | 0,69 | 0,4835 | 0,5579 | 0,7743 |
| letter | 0,6625 | 0,8888 | 0,6493 | FA | 0,7182 | 0,9066 | 0,9334 | 0,4828 | 0,8435 | 0,676 | 0,5257 | 0,6709 | 0,9450 |
| mnist | 0,7638 | 0,9484 | 0,8647 | 0,9189 | 0,4858 | 0,9318 | FA | 0,6151 | 0,9210 | 0,7604 | 0,2502 | 0,8540 | 0,9352 |
| optdigits | 0,8935 | 0,9991 | 0,8625 | 0,9846 | 0,4260 | 0,9983 | 0,9999 | 0,8105 | 0,8221 | 0,9086 | 0,6203 | 0,4751 | 0,9988 |
| satellite | 0,8630 | 0,8456 | 0,7834 | FA | 0,4745 | 0,8753 | 0,8740 | FA | 0,7957 | 0,7798 | 0,3099 | 0,7661 | 0,8517 |
| satimage-2 | 0,9836 | 0,9966 | 0,9910 | 0,9977 | 0,6745 | 0,9992 | 0,9826 | 0,6317 | 0,9967 | 0,9755 | 0,3968 | 0,9987 | 0,9993 |
| SpamBase | 0,8717 | 0,7132 | 0,8374 | 0,7730 | 0,3774 | 0,7036 | 0,6302 | 0,7377 | 0,8034 | 0,7807 | 0,4826 | 0,4579 | 0,8244 |
| speech | 0,6029 | 0,5075 | 0,5030 | 0,8741 | 0,4364 | 0,4853 | 0,4640 | 0,5138 | 0,5217 | 0,6076 | 0,4821 | 0,4553 | 0,5070 |
| SVHN | 0,6859 | 0,7192 | 0,5834 | 0,6989 | 0,5781 | 0,6788 | 0,6150 | 0,7055 | 0,6684 | 0,5894 | 0,4621 | 0,6076 | 0,6319 |
| Waveform | 0,8092 | 0,7530 | 0,6902 | 0,7115 | 0,5814 | 0,7623 | 0,5514 | 0,6049 | 0,5791 | 0,7214 | 0,7018 | 0,7223 | 0,7570 |
| WPBC | 0,6326 | 0,5695 | 0,5681 | 0,6156 | 0,5333 | 0,5830 | 0,5681 | 0,5972 | 0,5660 | 0,4907 | 0,4121 | 0,3355 | 0,4872 |
| Hepatitis | 0,6982 | 0,5030 | 0,6568 | 0,5207 | 0,2959 | 0,5680 | 0,4024 | FA | 0,7574 | 0,8284 | 0,3787 | 0,3905 | 0,7219 |
| MVTec-AD | 0,9806 | 0,9679 | 0,9755 | 0,9689 | 0,9662 | 0,9703 | 0,9645 | 0,6412 | 0,9776 | 0,7422 | 0,5179 | 0,9689 | 0,9727 |
| 20newsgroups | 0,5535 | 0,7854 | 0,6675 | FA | 0,7109 | 0,7260 | 0,6329 | 0,5313 | 0,8103 | 0,6063 | 0,4833 | 0,6715 | 0,7425 |

Table 6: Results of the Conover-Iman test for all the Deep methods.

| Method | AnoGAN | DIF | DeepSVDD | **GSAAL** | LUNAR | MO GAAL |
|---|---|---|---|---|---|---|
| AnoGAN | = | − − | − − | − − | − − | − − |
| DIF | ++ | = | − | − − | − − |  |
| DeepSVDD | ++ | + | = | − | − | ++ |
| **GSAAL** | ++ | ++ | + | = |  | ++ |
| LUNAR | ++ | ++ | + |  | = | ++ |
| MO GAAL | ++ |  | − − | − − | − − | = |

Table 7: Summary of the included components in the ablation study.

| Name | Subspace | Multiple $\mathcal{D}_i$ |
|---|---|---|
| GSAAL$_{✗✗}$ | ✗ | ✗ |
| GSAAL$_{✓✗}$ | ✓ | ✗ |
| GSAAL$_{✗✓}$ | ✗ | ✓ |
| **GSAAL** | ✓ | ✓ |

We will employ, once again, the Conover-Iman test to compare the performance of all configuration in a statistically sound way. Table 8 contains the results of the ablation experiment. As expected, our fully configured method significantly outperformed all of the others. This further confirms that the performance increase over our competitors comes directly from tackling the MV problem.

Table 8: Results of the Connover-Iman test for the ablation study.

|  | GSAAL$_{✗✗}$ | GSAAL$_{✓✗}$ | GSAAL$_{✗✓}$ | **GSAAL** |
|---|---|---|---|---|
| GSAAL$_{✗✗}$ | = | ++ | − − | − − |
| GSAAL$_{✓✗}$ | − − | = | − − | − − |
| GSAAL$_{✗✓}$ | ++ | ++ | = | − − |
| **GSAAL** | ++ | ++ | ++ | = |

