# OpenReview forum: "Generative Subspace Adversarial Active Learning for Outlier Detection in Multiple Views of High-dimensional Tabular Data"
_NeurIPS.cc/2024/Conference — Submitted to NeurIPS 2024_

### Official Review · Reviewer_vAno · 2024-07-11

**Soundness:** 4
**Presentation:** 4
**Contribution:** 4
**Rating:** 7
**Confidence:** 5

**Summary:**

This paper proposed GSAAL to simultaneously address three changeling problems in outlier detection: inlier assumption (IA), curse of dimensionality (CD), and multiple views (MV).

**Strengths:**

The paper has a good flow.
The paper proposed the first outlier detection method that explicitly addresses IA, CD, and MV simultaneously.
The paper has strong theoretical and empirical evidence to show the advancement of the proposed method.
The experimental design is solid and the numerous visual examples help to facilitate understanding.
The paper has good reproducibility with open codes.

**Weaknesses:**

some (but few) places to improve.

**Questions:**

1. There are a lot of abbreviations in this article. Before using them, they should be first defined.
2. In line 96, “Classical Methods” lacks recently published work, such as “Mean-shift outlier detection and filtering”.
3. In line 159, it reads “p_x(x) = p_{ux}(ux) for almost all x” (Equation 1). Please classify whether x here refers only to normal samples or also to outlier samples.
4. Why only K-NN-based baselines were shown in Fig. 3? How about other baselines? The font size of the text in Fig. 3 was too small.
5. Please explain the meaning of “FA” in Table 5.
6. The Y-axis in Figure 4 lacks a name.

**Limitations:**

The authors have analyzed the limitations sufficiently.

---

> ### Author Rebuttal · Authors · 2024-08-07
>
> - **There are a lot of abbreviations in this article. Before using them, they should be first defined.**
> 	- Thanks, we went through each abbreviation in the article and explained it accordingly.
> -  **In line 96, “Classical Methods” lacks recently published work, such as “Mean-shift outlier detection and filtering”.**
> 	- Thanks for the suggestion, we will add the reference in our related work.
> - **In line 159, it reads “$p_\mathbf{x}(x) = p_\mathbf{ux}(ux)$ for almost all $x$” (Equation 1). Please classify whether $x$ here refers only to normal samples or also to outlier samples.**
> 	- $𝑥$ here denotes a realization of $\mathbf{x}$, the inlier family. We will make this clearer by repeating the definitions for $𝑥$ and $𝑢$ to the camera-ready.
> -  **Why only K-NN-based baselines were shown in Fig. 3? How about other baselines? The font size of the text in Fig. 3 was too small.**
> 	- We compared with the quadratic runtime methods, as they were the best performing in the previous section. We have the results for all other methods but didn’t think that it was relevant to our claims. We will include the full figure in the camera ready to improve clearness, as well as fix the font size. The attached pdf in the *Author Rebuttal* contains the updated Figure.
> -  **Please explain the meaning of “FA” in Table 5.**
> 	-  Failure to Analyze. This might come due to different reasons, like the packages reporting a failure when taking the data (like OCSVM or ABOD). In the case of MO GAAL, it was because the network couldn’t converge in these data sets. We now added the explanation to the appendix.
> -  **The Y-axis in Figure 4 lacks a name.**
> 	- Thank you for the heads up. The figure is fixed and will be included in the camera-ready version of the paper

---

> > ### Comment · Reviewer_vAno · 2024-08-09
> > **Rebuttal review**
> >
> > I have read the rebuttal. Thanks for your explanation. Please adjust as you said.

---

### Official Review · Reviewer_FsUf · 2024-07-12

**Soundness:** 3
**Presentation:** 2
**Contribution:** 2
**Rating:** 4
**Confidence:** 4

**Summary:**

This paper presents this generalization of GAAL Generative Subspace Adversarial Active Learning (GSAAL) for outlier detection to address the limitation of the previous work such as multi-view and the curse of dimensionality, where the theoretical convergence, the scalability of the algorithm are discussed. Experiments on real dataset and synthetic tabular dataset are carried out to establish the validity of the approaches.

**Strengths:**

The manuscript presents a method called generative subspace adversarial active learning for outlier detection in multiple views. The proposed method called GSAAL provides the proof of convergence, the computation complexity and aims to address the curse of dimensionality. The outlier detection in high dimensional space indeed is an important and challenging solution. Thus, the proposed method can be a good solution to address this difficult problem.

The manuscript has compared the performance of GSAAL with other outlier detection approaches with detailed visual illustration and AUC. The experiments show advantages of the proposed solution over other competing methods. The experiments seems to be detailed.

**Weaknesses:**

The novelty of the work appears to be small. Theoretically, the derivation of theorem 1 is very similar to GAN derivation.

In this case, the paper needs to compare their solution both theoretically and experimentally with the related work for outlier detection using GAN such as [1] https://arxiv.org/pdf/1906.11632 such as AnoGAN, BioGAN and EGBAD.
[2] https://asp-eurasipjournals.springeropen.com/articles/10.1186/s13634-022-00943-7
If we compare the main equation (2) in the manuscript with the formulation in reference [1] with conditional GAN and BioGAN, it seems the main difference are the proposed method used multiple detectors and accumulated the performance, which should not be considered a large distinction.

Due to the lack of the comparison with generative adversarial network based approaches such as AnoGAN and EGBAD, the potential improvement of the purposed method against the state-of-the-art approaches is not clear. The novelty of the paper does not stand on the safe ground. The theoretical derivation is also similar to GAN derivation.

**Questions:**

Like mentioned above, in order to convince the readers, the paper should really clarify the different and improvement of their solution compared to GAN based solution and focus on explaining and justifying whether or why the improvement against GAN (if there are) is significant.

**Limitations:**

Limited innovation and lack of critical comparison with important reference are the main issues of the current manuscript.

---

> ### Author Rebuttal · Authors · 2024-08-07
>
> - **The novelty of the work appears to be small. Theoretically, the derivation of theorem 1 is very similar to GAN derivation.**
> 	- We do not agree that this can be a straightforward derivation from the classical GAN result. In GSAAL, as in GAAL methods, the detectors are trained after the generators using active learning (lines L179-L181). This makes GSAAL (and all GAAL methods) very different from a regular GAN by default, as established in [3]. If we agree that GSAAL does not provide enough novelty with respect to a GAN, we would also have to agree that the work of other GAAL methods, including [3], [12], [9], also lacks sufficient novelty. Next, comparing GSAAL to other GAAL methods, it is neither intuitive nor straightforward that using multiple detectors in subspaces will cause the network to learn the desired distribution. It is even more difficult to specify the conditions under which this would happen. In particular, without the theoretical formulation of multiple views, which is a cornerstone of our article, one cannot formulate Theorem 1. Furthermore, one also needs a Proposition, and the GAAL formulation highlighted in [9] and [3].
>
> - **In this case, the paper needs to compare their solution both theoretically and experimentally with the related work for outlier detection using GAN such as [1] [https://arxiv.org/pdf/1906.11632](https://arxiv.org/pdf/1906.11632) such as AnoGAN, BioGAN, and EGBAD. [2] [https://asp-eurasipjournals.springeropen.com/articles/10.1186/s13634-022-00943-7](https://asp-eurasipjournals.springeropen.com/articles/10.1186/s13634-022-00943-7)**
>
> 	- All models given as examples in the review focus mainly on image outlier detection, as highlighted in the survey provided by the reviewer (page 2 [1] : "*In the following sections, we present an analysis of the considered architecture. The term sample and image are used interchangeably since GANs can be used to detect anomalies on a wide range of domains, but all the analyzed architectures focused mostly on images*."). Since we are targeting tabular data, the focus is on tabular outlier detection methods (as mentioned in lines L 94 and L719-L720), including tabular data GAN-based models (GAAL-based). However, as we say in lines L225-L226, we have also considered outlier detection methods outside the tabular data domain, in particular, AnoGAN. Table 5 and Section B.3 show that AnoGAN performs worse than GSAAL on every single real data set out of 22, by a large margin. This is not surprising because of the domain change.
>
> - **If we compare the main equation (2) in the manuscript with the formulation in reference [1] with conditional GAN and BioGAN, it seems the main difference are the proposed method used multiple detectors and accumulated the performance, which should not be considered a large distinction.**
> 	- We do not agree that the similarity of two networks can be measured by the similarity of their loss functions. If one accepts this logic, then there is no difference between, say, a ResNet-18 [10] and a simple 2-layer MLP [11]. Similarly, GSAAL is completely different from BiGAN, Conditional GANs, and all the other methods listed.
>
> 		First, every single method mentioned by the reviewer uses some sort of reconstruction-based scoring function for the OD. 		GAAL-based methods do not rely on a reconstruction-based score, as they directly approximate the actual inlier density 		function [3]. To achieve this, these methods use active learning after convergence to make their detectors approximate such a 		function. Thus, GAAL-based methods (including our GSAAL) are fundamentally different from the other GAN-based 		approaches, as discussed in [3] and [9]. We have already mentioned these differences in lines L64-L67 and L130-L132.
>
> 		Furthermore, BiGAN trains a detector, a generator, and an encoder that learns the inverse of the generator $G^{-1}$. The detector then learns to classify the tuples given by $(x,E(x))$ and $(G(z),z)$, which changes the input space of the detector to the cartesian space $\mathcal{X}\times Z$.
> 		In contrast, GSAAL does not train an encoder and its detectors use $\mathcal{X}$ as input space.
>
> 		Conditional GANs, unlike BiGAN, use class labels $(x,y)$ instead of latent space representations. This makes them even more different from GSAAL, which does not use class labels, as they have a completely different setting. CGANs are meant for the Out of out-of-distribution detection setting, where there is a set of "normal" classes instead of just one.
>
> 		We will add these individual clarifications in the final manuscript.
>
> - **Like mentioned above, in order to convince the readers, the paper should really clarify the different and improvement of their solution compared to GAN based solution and focus on explaining and justifying whether or why the improvement against GAN (if there are) is significant.**
> 	- Our manuscript already mentions the differences to all other GAAL methods (GAN-based solutions in our field of work) in Section 3.2 lines L198-L204. The differences between GAAL methods and other GANs are summarized in lines L64-L67 and L130-L132; and presented in lines L177-L181, together with citations to more general GAAL work that delves deeply into these differences [12], [9], [3]. To justify our increase based on these changes, we performed an ablation study (as stated in line L224). There we significantly outperform the reduced networks (see Table 8). We will make our improvements over GAN clearer as part of the contributions in Section 1 and Section 5.2.

---

> > ### Comment · Reviewer_FsUf · 2024-08-10
> > **Response to the rebuttal**
> >
> > Thanks for the detailed comments.
> >
> > After reading other reviewer's comments and the rebuttal, I decide to main my rating.

---

> > > ### Author Response · Authors · 2024-08-12
> > > **Response**
> > >
> > > We are sorry, but this does not help us much to improve our paper. Could you please be more specific about which of our arguments or clarifications you find unconvincing and why? In our rebuttal, we showed that the methods requested in comment 2 are already part of our experiments, the information from comments 1 and 4 is included, and the methods in comment 3 are not proposed for tabular data or even applicable, as noted in our paper and cited references.

---

### Official Review · Reviewer_hp34 · 2024-07-12

**Soundness:** 2
**Presentation:** 1
**Contribution:** 2
**Rating:** 4
**Confidence:** 3

**Summary:**

The main contribution of this paper is to improve existing work on Generative Adversarial Active Learning (GAAL) by using multiple discriminators for multiple views to detect outliers in tabular data. The training mechanism is similar to existing works. The paper also introduces a theoretical analysis on Multiple Views (MV). As claimed by the authors, GAAL addressed the problems of Inlier Assumption (IA) and Curse of Dimensionality (CD), but missed Multiple Views (MV), which is the main focus of this paper. The experimental results compare the proposed method to GAAL and some other classical methods such as OCSVM and KNN, ....

**Strengths:**

The paper introduces an interesting view about MV and proposes a new method to address this MV problem together with theoretical analysis.

**Weaknesses:**

* The empirical results are not strong (or at least unclear in the way the authors presented in the main paper); most of the experiments are on synthetic datasets.
* The results on the real dataset do not seem to show significant improvements compared to existing work (or at least it is hard to observe this when reading the paper). Perhaps the authors could improve the writing and highlight the results better. It is unclear to me why the experiments on the real dataset were put in the Appendix, as it is an important result.
* The paper claims at the beginning that it not only improves the MV problem but also the IA and CD problems, but this is hard to see with the current writing of the paper. Could the authors highlight the experiments in the paper to prove that claim?

**Questions:**

1. How do you define the number of discriminators and the number of realizations u?
2. In Fig. 3, why is GAAL missing? The paper claims it is faster in terms of time complexity. Which results show this?

---

> ### Author Rebuttal · Authors · 2024-08-07
>
> - **The empirical results are not strong (or at least unclear in the way the authors presented them in the main paper); most of the experiments are on synthetic datasets.**
> 	- We run experiments on 22 datasets, which are more datasets than relevant and popular competitors in the field of OD [3],[4],[13],[5] use. Our results show that GSAAL is statistically significantly better than the majority of competitors, including the most related GAN-based [3]. Additionally, we also have more experiments utilizing real data in the Appendix. To show that GSAAL has much better scalability than conventional methods, which were not significantly worse in real data experiments, we use synthetic data. This allows us to control the number of features and samples. In addition, we verified that the high performance of GSAAL is due to its ability to handle MV. The only way to measure and visualize this effect is with synthetic data. We mentioned this and commented on the data generation process in Section 4.2, lines L263--L268. We ask the reviewer to explain which results are not convincingly strong and why, otherwise we cannot improve based on this information.
>
> - **The results on the real dataset do not seem to show significant improvements compared to existing work (or at least it is hard to observe this when reading the paper). Perhaps the authors could improve the writing and highlight the results better. It is unclear to me why the experiments on the real dataset were put in the Appendix, as it is an important result.**
> 	- Table 3 presents a statistical summary of the results, as this is the standard way to compare OD methods, see [6],[3],[7],[8]. This summary contains the results of the pairwise statistical tests (as explained in lines L297-L299) and shows the statistical significance of the improvements, as requested by the question. We analyze this table further in lines L300-L303 and explain why it shows the superiority of GSAAL. If the reviewer disagrees with the significance of this particular test, we would appreciate specific reasoning. As it stands, we cannot respond adequately without more detailed feedback. The statement that the experiments on real datasets are in the Appendix is largely incorrect. As mentioned in section *4.3.1 Real-World Performance*, we used real datasets for the experiments, reported the statistical test results, and analyzed them. The raw AUC results are included in the Appendix (as stated in line 289) because it is common practice to maintain a proper flow of text. In addition, we considered both the time and MV experiments more important to include in the main body due to the page limit.
>
> - **The paper claims at the beginning that it not only improves the MV problem but also the IA and CD problems, but this is hard to see with the current writing of the paper. Could the authors highlight the experiments in the paper to prove that claim?**
> 	- We thank the reviewer for this point and have clarified it in the updated manuscript. Indeed, GSAAL addresses MV, IA, and CD. This is by design since GSAAL is a member of the GAAL family of methods that already satisfy IA and CD [3]. Therefore, this paper focuses on MV. Nevertheless, we agree with the reviewer that IA and CD should be verified. Due to this, we have IA experiments in the appendix, as mentioned in line 223. To account for CD, we have used real-world high-dimensional datasets (such as CIFAR, SVNH, 20news, F-MNIST, MNIST, and MVTec among others from [7]), as also done by [3], [5], [4] and [7]  with the same purpose.
>
> - **How do you define the number of discriminators and the number of realizations u?**
> 	-  We choose *$k = 2\sqrt(d)$* detectors for the experiments in Section 4.3.1, as indicated in line 242. We also study our approach with respect to the number of discriminators; see Appendix section *B.4.Parameter Sensitivity* and line 223. Since each detector is fitted in a unique subspace, the number of detectors is equal to the number of realizations of $\mathbf{u}$.
>
> -  **In Fig. 3, why is GAAL missing? The paper claims it is faster in terms of time complexity. Which results show this?**
> 	- We did not intend and, hopefully, never made the statement that GSAAL is faster than MO-GAAL. Please point to the exact place in the text otherwise so that we can remove it. We only included the quadratic time competitors because we claimed to have a better inference time complexity (Section 3.3). We have added the remaining methods into the plot which is now included in the manuscript. We have no new insights regarding the plot. Fig. 3 is included in the attached global document.

---

### Author Rebuttal · Authors · 2024-08-07

We thank all of the reviewers for their efforts and their time. Our detailed response is in each individual message. The attached pdf contains edits to Fig.3 and Fig.4 and new information in Fig.3 as requested in the reviews. The new information does not change any of our conclusions. The references of all citations in the comments are included in the following.

REFERENCES

[1]  Federico Di Mattia, Paolo Galeone, Michele De Simoni, & Emanuele Ghelfi. (2021). A Survey on GANs for Anomaly Detection.

[2] Luo, X., Jiang, Y., Wang, E. _et al._ Anomaly detection by using a combination of generative adversarial networks and convolutional autoencoders. _EURASIP J. Adv. Signal Process._**2022**, 112 (2022). https://doi.org/10.1186/s13634-022-00943-7

[3] Y. Liu, Z. Li, C. Zhou, Y. Jiang, J. Sun, M. Wang, and X. He. Generative adversarial active learning for unsupervised outlier detection. IEEE Transactions on Knowledge and Data Engineering, 32(8):1517–1528, 2020.

[4] H. Xu, G. Pang, Y. Wang, and Y. Wang. Deep isolation forest for anomaly detection. IEEE Transactions on Knowledge and Data Engineering, 35(12):12591–12604, 2023.

[5] L. Ruff, R. Vandermeulen, N. Goernitz, L. Deecke, S. A. Siddiqui, A. Binder, E. Müller, and M. Kloft. Deep one-class classification. In J. Dy and A. Krause, editors, Proceedings of the 35th International Conference on Machine Learning, volume 80 of Proceedings of Machine Learning Research, pages 4393–4402. PMLR, 10–15 Jul 2018.

[6] G. O. Campos, A. Zimek, J. Sander, R. J. G. B. Campello, B. Micenková, E. Schubert, I. Assent, and M. E. Houle. On the evaluation of unsupervised outlier detection: measures, datasets, and an empirical study. Data Mining and Knowledge Discovery, 30(4):891–927, Jul 2016.

[7] S. Han, X. Hu, H. Huang, M. Jiang, and Y. Zhao. Adbench: Anomaly detection benchmark. In S. Koyejo, S. Mohamed, A. Agarwal, D. Belgrave, K. Cho, and A. Oh, editors, Advances in Neural Information Processing Systems, volume 35, pages 32142–32159. Curran Associates, Inc., 2022.

[8] A. Goodge, B. Hooi, S.-K. Ng, and W. S. Ng. Lunar: Unifying local outlier detection methods via graph neural networks. ArXiv, abs/2112.05355, 2021.

[9] J.-J. Zhu and J. Bento. Generative adversarial active learning. arXiv preprint arXiv:1702.07956, 2017.

[10]  K. He, X. Zhang, S. Ren and J. Sun, "Deep Residual Learning for Image Recognition," 2016 IEEE Conference on Computer Vision and Pattern Recognition (CVPR), Las Vegas, NV, USA, 2016, pp. 770-778, doi: 10.1109/CVPR.2016.90. keywords: {Training;Degradation;Complexity theory;Image recognition;Neural networks;Visualization;Image segmentation},

[11] I. Goodfellow, Y. Bengio, and A. Courville. Deep Learning. MIT Press, 2016. http: //www.deeplearningbook.org.

[12] J. Guo, Z. Pang, M. Bai, P. Xie, and Y. Chen. Dual generative adversarial active learning. Applied Intelligence, 51(8):5953–5964, Aug 2021.

[13] T. Schlegl, P. Seeböck, S. M. Waldstein, U. Schmidt-Erfurth, and G. Langs. Unsupervised anomaly detection with generative adversarial networks to guide marker discovery. In M. Ni- ethammer, M. Styner, S. Aylward, H. Zhu, I. Oguz, P.-T. Yap, and D. Shen, editors, Information Processing in Medical Imaging, pages 146–157, Cham, 2017. Springer International Publishing.

---

### Comment · Area_Chair_68zF · 2024-08-14

Dear all,

Thanks for your time and efforts in reviewing this paper. This is the right and emerging time to discuss this paper with the authors.

The authors provided their rebuttal, and some reviewers have posted their partial discussion about this paper.

Any discussion is welcome and you may consider reading each others' reviews, posting a question, and reaching a consensus.

Best,
Your AC

---

### Decision · Program_Chairs · 2024-09-25

**Decision:**

Reject

**Comment:**

This paper provides a mathematical formulation of multiple views, convergence guarantees for the discriminators, and scalability results for Generative Subspace Adversarial Active Learning. Two of the three reviewers were inclined to reject this manuscript, mainly because it lacked sufficient innovation and strong experiments. After discussion, the rebuttal failed to address these concerns, and the final decision is reject.